# CROSS DOMAIN IMITATION LEARNING

## ABSTRACT

We study the question of how to imitate tasks across domains with discrepancies such as embodiment and viewpoint mismatch. Many prior works require paired, aligned demonstrations and an additional RL procedure for the task. However, paired, aligned demonstrations are seldom obtainable and RL procedures are expensive. In this work, we formalize the Cross Domain Imitation Learning (CDIL) problem, which encompasses imitation learning in the presence of viewpoint and embodiment mismatch. Informally, CDIL is the process of learning how to perform a task optimally, given demonstrations of the task in a distinct domain. We propose a two step approach to CDIL: alignment followed by adaptation. In the alignment step we execute a novel unsupervised MDP alignment algorithm, Generative Adversarial MDP Alignment (GAMA), to learn state and action correspondences from *unpaired, unaligned* demonstrations. In the adaptation step we leverage the correspondences to zero-shot imitate tasks across domains. To describe when CDIL is feasible via alignment and adaptation, we introduce a theory of MDP alignability. We experimentally evaluate GAMA against baselines in both embodiment and viewpoint mismatch scenarios where aligned demonstrations don't exist and show the effectiveness of our approach.

## 1 INTRODUCTION

Humans possess an astonishing ability to recognize latent structural similarities between behaviors in related but distinct domains, and learn new skills from cross domain demonstrations alone. Not only are we capable of learning from third person observations that have no obvious correspondence to our internal self representations (Stadie et al., 2017; Liu et al., 2018; Sermanet et al., 2018), but we also are capable of imitating agents with different embodiments (Gupta et al., 2017; Rizzolatti & Craighero, 2004) as can be observed in an infant's learning of visuomotor skills from adults with different biomechanics and physical capabilities (Jones, 2009). Previous work in neuroscience (Marshall & Meltzoff, 2015) and robotics (Kuniyoshi & Inoue, 1993; Kuniyoshi et al., 1994) have recognized the pitfalls of exact behavioral cloning in the presence of domain discrepancies and posited that the effectiveness of the human imitation learning mechanism hinges, crucially, on the capability to learn structure preserving domain correspondences. These correspondences enable the learner to internalize the expert demonstrations and produce a reconstruction of the behavior in the self domain. Consider a young child that has learned to associate his internal body map with the limbs of an adult. When the adult demonstrates running, the child is able to imagine himself running, and reproduce the behavior.

Recently, separate solutions have been proposed for imitation learning across two kinds of domain discrepancies: embodiment (Gupta et al., 2017) and viewpoint (Liu et al., 2018; Sermanet et al., 2018) mismatch. These works (Liu et al., 2018; Sermanet et al., 2018; Gupta et al., 2017) require *paired, time-aligned demonstrations* to obtain state correspondences and an extra RL step with a proxy reward. However, paired, aligned demonstrations are seldom obtainable and RL loops are expensive. In this work we formalize the Cross Domain Imitation Learning (CDIL) problem which encompasses prior work in imitation learning across domains with viewpoint and embodiment mismatch. Informally, CDIL is the *process of learning how to perform a task optimally in a self domain, given demonstrations of the task in a distinct expert domain*. We propose a two-step approach to CDIL: alignment followed by adaptation. In the alignment step we execute a novel unsupervised MDP alignment algorithm, Generative Adversarial MDP Alignment (GAMA), to learn state, action maps from *unpaired, unaligned* demonstrations. In the adaptation step we leverage the learned state, action maps to zero-shot imitate tasks across domains without an additional RL step. To shed light on when CDIL can be solved by alignment and adaptation, we first introduce a class of structure preserving maps, called MDP reductions, that adapts optimal policies between MDPs (section 3). We further characterize a family of MDP pairs that share reductions, formally state the MDP alignment problem, and elucidate its connection to CDIL. In section 4, 5 we derive GAMA,

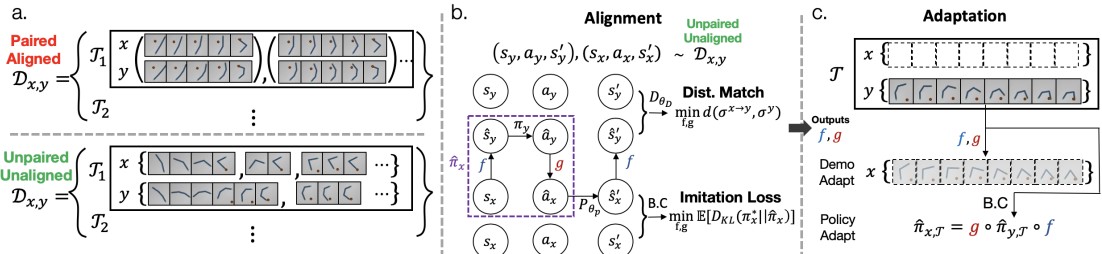

Figure 1: (a). Illustration of paired, aligned vs unpaired, unaligned demonstrations in the alignment task set $\mathcal{D}_{x,y}$ (b). Alignment: we learn state, action maps $f, g$ between the self ($x$) and expert ($y$) domain from unpaired, unaligned demonstrations by minimizing a distribution matching loss and an imitation loss. (c) Adaptation: adapt the expert domain policy $\pi_{y,\mathcal{T}}$ or demonstrations to obtain a self domain policy $\hat{\pi}_{x,\mathcal{T}}$

a simple training algorithm to learn MDP reductions. In section 6, we experimentally evaluate GAMA and find that meaningful state correspondences between various domains are learned from unpaired, unaligned demonstrations. We then compare the CDIL performance of GAMA against several baselines in both embodiment and viewpoint mismatch scenarios and show the effectiveness of our approach.

## 2    CROSS DOMAIN IMITATION LEARNING PROBLEM STATEMENT

An infinite horizon Markov Decision Process (MDP) $\mathcal{M} \in \Omega$ with deterministic dynamics is a tuple $(\mathcal{S}, \mathcal{A}, P, \eta, R)$ where $\Omega$ is the set of all MDPs, $\mathcal{S}$ is the state space, $\mathcal{A}$ is the action space, $P : \mathcal{S} \times \mathcal{A} \to \mathcal{S}$ is a (deterministic) transition function, $R : \mathcal{S} \times \mathcal{A} \to \mathbb{R}$ is the reward function, and $\eta$ is the initial state distribution. A domain is an MDP without the reward, i.e $(\mathcal{S}, \mathcal{A}, P, \eta)$. Intuitively, a domain fully characterizes the embodied agent and the environment dynamics, but not the desired behavior. A task $\mathcal{T}$ is a label for an MDP corresponding to the high level description of optimal behavior, such as "walking". $\mathcal{T}$ is analogous to category labels for images. An MDP with domain $x$ for task $\mathcal{T}$ is denoted by $\mathcal{M}_{x,\mathcal{T}} = (\mathcal{S}_x, \mathcal{A}_x, P_x, \eta_x, R_{x,\mathcal{T}})$, where $R_{x,\mathcal{T}}$ is a reward function encapsulating the behavior labeled by $\mathcal{T}$. For example, different reward functions are needed to realize the "walking" behavior in two morphologically different humanoids. A (stationary) policy for $\mathcal{M}_{x,\mathcal{T}}$ is a map $\pi_{x,\mathcal{T}} : \mathcal{S}_x \to \mathcal{B}(\mathcal{A}_x)$ where $\mathcal{B}$ is the set of probability measures on $\mathcal{A}_x$ and an optimal policy $\pi^*_{x,\mathcal{T}} = \arg\max_{\pi_x} J(\pi_x)$ achieves the highest policy performance $J(\pi_x) = \mathbb{E}_{\pi_x}[\sum_{t=0}^{\infty} \gamma^t R_{x,\mathcal{T}}(s_x^{(t)}, a_x^{(t)})]$ where $0 < \gamma < 1$ is a discount factor. A demonstration of length $H$ is a sequence of state, action tuples $\tau_{\mathcal{M}_{x,\mathcal{T}}} = \{(s_x^{(t)}, a_x^{(t)})\}_{t=1}^{H}$ sampled from an optimal policy and $\mathcal{D}_{\mathcal{M}_{x,\mathcal{T}}} = \{\tau_{\mathcal{M}_{x,\mathcal{T}}}^{(k)}\}_{k=1}^{K}$ is a set of demonstrations for $\mathcal{M}_{x,\mathcal{T}}$

Let $\mathcal{M}_{x,\mathcal{T}}, \mathcal{M}_{y,\mathcal{T}}$ be self and expert MDPs for a target task $\mathcal{T}$. Given expert domain demonstrations $\mathcal{D}_{\mathcal{M}_{y,\mathcal{T}}}$, Cross Domain Imitation Learning (CDIL) aims to determine an optimal self domain policy $\pi^*_{x,\mathcal{T}}$ without access to the reward function $R_{x,\mathcal{T}}$. In this work we propose to first solve an MDP alignment problem and then leverage the alignments to zero-shot imitate expert domain demonstrations. Like prior work (Gupta et al., 2017; Liu et al., 2018; Sermanet et al., 2018), we assume the availability of an alignment task set $\mathcal{D}_{x,y} = \{(\mathcal{D}_{\mathcal{M}_{x,\mathcal{T}_i}}, \mathcal{D}_{\mathcal{M}_{y,\mathcal{T}_i}})\}_{i=1}^{N}$ containing demonstrations for $N$ tasks $\{\mathcal{T}_i\}_{i=1}^{N}$ from both the self and expert domain. $\mathcal{D}_{x,y}$ could, for example, contain both robot ($x$) and human ($y$) demonstrations for a set primitive tasks such as walking, running, and jumping. Unlike prior work, demonstrations are *unpaired and unaligned*, i.e $(s_x^{(t)}, s_y^{(t)})$ may not be a valid state correspondence. (see Figure 1(a)) Paired, time-aligned cross domain data is expensive and may not even exist when task execution rates differ or there exists systematic embodiment mismatch between the domains. For example, a child can imitate an adult running, but not achieve the same speed. Our set up emulates a natural setting in which humans compare how they perform tasks to how other agents perform the same tasks in order to find structural similarities and identify domain correspondences. We now proceed to introduce a theoretical framework that explains how and when the CDIL problem can be solved by MDP alignment followed by adaptation.

## 3    ALIGNABLE MDPS

Let $\Pi^*_{\mathcal{M}}$ be the set of all optimal policies for MDP $\mathcal{M}$. We define an occupancy measure (Syed et al., 2008) $q_\pi : \mathcal{S} \times \mathcal{A} \to \mathbb{R}$ for policy $\pi$ as $q_\pi(s, a) = \pi(a|s) \sum_{t=0}^{\infty} \gamma^t \Pr(s^{(t)} = s; \pi, P, \eta)$.

**Definition 1.** *An optimality function* $O_{\mathcal{M}_x} : \mathcal{S}_x \times \mathcal{A}_x \to \{0, 1\}$ *for an MDP* $\mathcal{M}_x$ *satisfies:* $O_{\mathcal{M}_x}(s_x, a_x) = 1$ *if* $\exists \pi^*_x \in \Pi^*_{\mathcal{M}_x}$ *such that* $(s_x, a_x) \in \text{supp}(q_{\pi^*_x})$ *and* $O_{\mathcal{M}_x}(s_x, a_x) = 0$ *otherwise.*

**Definition 2.** *An **MDP reduction** from $\mathcal{M}_x = (\mathcal{S}_x, \mathcal{A}_x, P_x, \eta_x, R_x)$ to $\mathcal{M}_y = (\mathcal{S}_y, \mathcal{A}_y, P_y, \eta_y, R_y)$ is a tuple $r = (\phi, \psi)$ where $\phi : \mathcal{S}_x \to \mathcal{S}_y, \psi : \mathcal{A}_x \to \mathcal{A}_y$ are maps that preserve:*

*1. (optimal policy) $\forall s_x \in \mathcal{S}_x, a_x \in \mathcal{A}_x, s_y \in \mathcal{S}_y, a_y \in \mathcal{A}_y,$*

$$O_{\mathcal{M}_y}(\phi(s_x), \psi(a_x)) = 1 \quad \Rightarrow \quad O_{\mathcal{M}_x}(s_x, a_x) = 1 \tag{1}$$

$$O_{\mathcal{M}_y}(s_y, a_y) = 1 \quad \Rightarrow \quad \phi^{-1}(s_y) \neq \emptyset, \psi^{-1}(a_y) \neq \emptyset \tag{2}$$

*2. (dynamics) $\forall s_y, s'_y \in \mathcal{S}_y, a_y \in \mathcal{A}_y$ where $O_{\mathcal{M}_y}(s_y, a_y) = 1,$*

$$P_y(s_y, a_y) = \phi(P_x(s_x, a_x)) \quad \forall s_x \in \phi^{-1}(s_y), a_x \in \psi^{-1}(a_y) \tag{3}$$

*where we define $\phi^{-1}(s_y) = \{s_x | \phi(s_x) = s_y\}$, $\psi^{-1}(a_y) = \{a_x | \psi(a_x) = a_y\}$. Furthermore, r is an **MDP permutation** if and only if $\phi, \psi$ are bijective maps.*

In words, Eq. 1 states that only optimal state, action pairs in $x$ can be mapped to optimal state, action pairs in $y$ and Eq. 2 states that $r$ must be surjective on the set of optimal state, action pairs in $y$. Eq. 3 states that a reduction must preserve (deterministic) dynamics. We use the notation $\mathcal{M}_x \geq_{\phi,\psi} \mathcal{M}_y$ to denote that $(\phi, \psi)$ is a reduction from $\mathcal{M}_x$ to $\mathcal{M}_y$, and the shorthand $\mathcal{M}_x \geq \mathcal{M}_y$ to denote that $\mathcal{M}_x$ reduces to $\mathcal{M}_y$. To gain an intuitive understanding of MDP reductions, picture the execution trace of an optimal policy as a directed

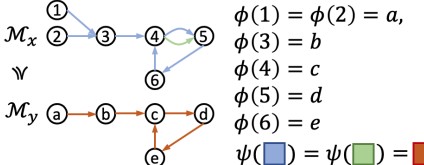

$\phi(1) = \phi(2) = a,$
$\phi(3) = b$
$\phi(4) = c$
$\phi(5) = d$
$\phi(6) = e$
$\psi(\blacksquare) = \psi(\blacksquare) = \blacksquare$

Figure 2: Example MDP reduction from $\mathcal{M}_x$ to $\mathcal{M}_y$. $\phi, \psi$ are state and action maps

graph with colored edges in which the nodes correspond to states visited by an optimal policy, and the colored edges correspond to actions taken. An MDP reduction from $\mathcal{M}_x$ to $\mathcal{M}_y$ homomorphs the execution graph of an optimal policy in $\mathcal{M}_x$ to a execution graph of an optimal policy in $\mathcal{M}_y$. Figure 2 shows an example of a valid reduction from $\mathcal{M}_x$ to $\mathcal{M}_y$: states $1, 2$ in $\mathcal{S}_x$ are mapped (merged) to state $a$ in $\mathcal{S}_y$ and the blue, green actions in $\mathcal{A}_x$ are mapped to the brown action in $\mathcal{A}_y$. Intuitively, if $\mathcal{M}_x \geq_{\phi,\psi} \mathcal{M}_y$, then $(\phi, \psi)$ compresses $\mathcal{M}_x$ by merging all optimal state, action pairs that have identical dynamics properties.

**Definition 3.** *Two MDPs $\mathcal{M}_x, \mathcal{M}_y$ are **alignable** if and only if $\mathcal{M}_x \geq \mathcal{M}_y$ or $\mathcal{M}_y \geq \mathcal{M}_x$.*

Definition 3 states that MDPs are alignable if reductions exists between them, meaning that they share structure. We use $\Gamma(\mathcal{M}_x, \mathcal{M}_y) = \{(\phi, \psi) | \mathcal{M}_x \geq_{\phi,\psi} \mathcal{M}_y\}$ to denote the set of all valid reductions from $\mathcal{M}_x$ to $\mathcal{M}_y$. Reductions have a particularly useful property which is that they adapt policies across alignable MDPs. Consider a state map $f : \mathcal{S}_x \to \mathcal{S}_y$, an inverse action map $g : \mathcal{A}_y \to \mathcal{A}_x$, and a composite policy $\hat{\pi}_x = g \circ \pi_y \circ f$ (see Figure 1(b)). In words, $\hat{\pi}_x$ maps a self state to an expert state via $f$, simulates the expert's action choice for the mapped state via $\pi_y$, then chooses a self action that corresponds to the simulated expert action with $g$. The following lemma holds for $\hat{\pi}_x$.

**Lemma 1.** *Let $\mathcal{M}_x, \mathcal{M}_y$ be MDPs satisfying Assumption 1 (see Supp. Materials), $\mathcal{M}_x \geq_{\phi,\psi} \mathcal{M}_y$, and $\pi_y$ be optimal in $\mathcal{M}_y$. $\forall g : \mathcal{A}_y \to \mathcal{A}_x$ s.t $\psi \circ g(a_y) = a_y$ $\forall a_y \in \{a_y | \exists s_y \in \mathcal{S}_y \text{ s.t } O_{\mathcal{M}_y}(s_y, a_y) = 1\}$, it holds that $\hat{\pi}_x = g \circ \pi_y \circ \phi$ is optimal in $\mathcal{M}_x$.*

Lemma 1 states that the state, action maps $(f, g^{-1})$ chosen to be a reduction can adapt optimal policies between alignable MDPs. Here onwards we interchangeably refer to $(f, g)$ as "alignments". We now show how the CDIL problem can be solved by first solving an MDP alignment problem followed by an adaptation step.

**Definition 4.** *Let $(\mathcal{M}_x, \mathcal{M}_y), (\mathcal{M}_x', \mathcal{M}_y') \in \Omega^2$ be two MDP pairs. Then, $(\mathcal{M}_x, \mathcal{M}_y) \sim (\mathcal{M}_x', \mathcal{M}_y')$, i.e they are **joint alignable**, if and only if $\Gamma(\mathcal{M}_x, \mathcal{M}_y) \cap \Gamma(\mathcal{M}_x', \mathcal{M}_y') \neq \emptyset$.*

In words, two MDP pairs are joint alignable if there exists a shared reduction. We define an equivalence class $[(\mathcal{M}_x, \mathcal{M}_y)]_\sim = \{(\mathcal{M}_x', \mathcal{M}_y') \mid (\mathcal{M}_x', \mathcal{M}_y') \sim (\mathcal{M}_x, \mathcal{M}_y)\}$ of MDP pairs that share reductions. Overloading notation, $\Gamma(\{(\mathcal{M}_x^i, \mathcal{M}_y^i)\}_{i=1}^N) = \{(\phi, \psi) \mid (\phi, \psi) \in \Gamma(\mathcal{M}_x^1, \mathcal{M}_x^1) \cap \cdots \cap \Gamma(\mathcal{M}_x^N, \mathcal{M}_x^N)\}$. We now formally state the MDP alignment problem: Let $(\mathcal{M}_{x,\mathcal{T}}, \mathcal{M}_{y,\mathcal{T}})$ be an MDP pair for a target task $\mathcal{T}$. Given an alignment task set $\mathcal{D}_{x,y} = \{(\mathcal{D}_{\mathcal{M}_{x,\mathcal{T}_i}}, \mathcal{D}_{\mathcal{M}_{y,\mathcal{T}_i}})\}_{i=1}^N$ comprising unpaired, unaligned demonstrations for MDP pairs $\{(\mathcal{M}_{x,\mathcal{T}_i}, \mathcal{M}_{y,\mathcal{T}_i})\}_{i=1}^N \subseteq [(\mathcal{M}_{x,\mathcal{T}}, \mathcal{M}_{y,\mathcal{T}})]_\sim$, determine $(\phi, \psi) \in \Gamma(\{(\mathcal{M}_{x,\mathcal{T}_i}, \mathcal{M}_{y,\mathcal{T}_i})\}_{i=1}^N)$ such that $(\phi, \psi) \in \Gamma(\mathcal{M}_{x,\mathcal{T}}, \mathcal{M}_{y,\mathcal{T}})$. As shown in Figure 3, with more MDP pairs, there are likely a smaller the number of joint alignments $|\Gamma(\{(\mathcal{M}_{x,\mathcal{T}_i}, \mathcal{M}_{y,\mathcal{T}_i})\}_{i=1}^N)|$ and, as a result, $(\phi, \psi) \in \Gamma(\{(\mathcal{M}_{x,\mathcal{T}_i}, \mathcal{M}_{y,\mathcal{T}_i})\}_{i=1}^N)$ is more likely to "generalize" to an MDP pair for a new target task $(\mathcal{M}_{x,\mathcal{T}}, \mathcal{M}_{y,\mathcal{T}})$ in the equivalence class. Analogously, in a standard supervised learning problem, more training data is likely to shrink the set of models performing optimally on the training set but poorly on the test set.

We can then use $(\phi, \psi)$ for CDIL: given cross domain demonstrations $\mathcal{D}_{\mathcal{M}_y, \mathcal{T}}$ for the target task $\mathcal{T}$, learn an expert domain policy $\pi_{y, \mathcal{T}}$, and adapt it into the self domain using $(\phi, \psi)$ according to Lemma 1.

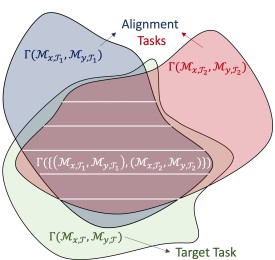

We can now assess when domains with embodiment and viewpoint mismatch have meaningful state correspondences, i.e MDP reductions, thus allowing for cross domain imitation. The states of a human expert with more degrees of freedom than a robot imitator can be merged into the robot states if the task only requires the robot's degrees of freedom and the execution traces share structure, e.g traces are both cycles. However, if the task requires all degrees of freedom possessed only by the human, the robot cannot find meaningful correspondences, and also cannot imitate the task. Two MDPs

Figure 3: Illustration of MDP alignment problem

for different viewpoints of an agent performing a task are MDP permutations since there is a one-to-one correspondence between state, actions at same timestep in the execution trace of an optimal policy.

## 4 LEARNING MDP REDUCTIONS

We now derive objectives that can be optimized to learn MDP reductions. We propose distribution matching and policy performance maximization. We first define the distributions to be matched.

**Definition 5.** *Let $\mathcal{M}_x$, $\mathcal{M}_y$ be two MDPs and $\hat{\pi}_x = g \circ \pi_y \circ f$ for $f : \mathcal{S}_x \to \mathcal{S}_y, g : \mathcal{A}_y \to \mathcal{A}_x$ and policy $\pi_y$. $\mathcal{P} = \{\hat{s}_y^{(t)}, \hat{a}_y^{(t)}\}_{t \geq 0}$ is the **co-domain policy execution process** realized by running $\hat{\pi}_x$, i.e:*

$$s_x^{(0)} \sim \eta_x, \hat{s}_y^{(t)} = f(s_x^{(t)}), \hat{a}_y^{(t)} \sim \pi_y(\cdot | \hat{s}_y^{(t)}), a_x^{(t)} = g(\hat{a}_y^{(t)}), s_x^{(t+1)} = P_x(s_x^{(t)}, a_x^{(t)}) \quad \forall t \geq 0 \quad (4)$$

The target distribution $\sigma_{\pi_y}^y$ is over transitions uniformly sampled from execution traces of $\pi_y$ and the proxy distribution $\sigma_{\hat{\pi}_x}^{x \to y}$ is over cross domain transitions uniformly sampled from realizations of $\mathcal{P}$.

$$\sigma_{\pi_y}^y(s_y, a_y, s_y') = \lim_{T \to \infty} \frac{1}{T} \sum_{t=0}^{T-1} \Pr(s_y^{(t)} = s_y, a_y^{(t)} = a_y, s_y^{(t+1)} = s_y'; \pi_y, P_y, \eta_y) \quad (5)$$

$$\sigma_{\hat{\pi}_x}^{x \to y}(s_y, a_y, s_y') = \lim_{T \to \infty} \frac{1}{T} \sum_{t=0}^{T-1} \Pr(\hat{s}_y^{(t)} = s_y, \hat{a}_y^{(t)} = a_y, \hat{s}_y^{(t+1)} = s_y'; \mathcal{P}) \quad (6)$$

We now propose three concrete objectives: 1. $\hat{\pi}_x$ is optimal, 2. $\sigma_{\hat{\pi}_x}^{x \to y} = \sigma_{\pi_y}^y$, 3. $g$ is injective. In other words, we seek to learn $f, g$ that matches distributions over transition tuples in domain $y$ while maximizing policy performance in domain $x$. The former captures the dynamics preservation property from Eq. 3 and the latter captures the optimal policy preservation property from Eq. 1, 2. The following theorem uncovers the connection between our objectives and MDP reductions.

**Theorem 1.** *Let $\mathcal{M}_x, \mathcal{M}_y$ be MDPs satisfying Assumption 1 (see Supp Materials). If $\mathcal{M}_x \geq \mathcal{M}_y$, then $\exists f : \mathcal{S}_x \to \mathcal{S}_y, g : \mathcal{A}_y \to \mathcal{A}_x$, and an optimal covering policy $\pi_y$ (Supp Materials, Def 6) that satisfy objectives 1, 2. Conversely, if $\exists f : \mathcal{S}_x \to \mathcal{S}_y, g : \mathcal{A}_y \to \mathcal{A}_x$ and an optimal covering policy $\pi_y$ satisfying objectives 1, 2, 3, then $\mathcal{M}_x \geq \mathcal{M}_y$ and $\exists (\phi, \psi) \in \Gamma(\mathcal{M}_x, \mathcal{M}_y)$ s.t $f = \phi$ and $\psi \circ g(a_y) = a_y, \forall a_y \in \mathcal{A}_y$.*

Theorem 1 states that if two MDP are alignable, then objectives 1, 2 can be satisfied. Conversely, if objectives 1, 2, 3 can be satisfied for two MDPs, then they must be alignable and all solutions $(f, g)$ are MDP reductions. While Theorem 1 requires alignable MDPs to guarantee identifiability, our experiments will also run on MDPs that are not perfectly alignable, i.e. Eq. 1, 2, 3 do not hold exactly, but intuitively share structure. In the next section, we propose a simple algorithm to learn MDP reductions.

## 5 GENERATIVE ADVERSARIAL MDP ALIGNMENT

Building on Theorem 1, we propose the following general form training objective for aligning MDPs:

$$\min_{f, g} -J(\hat{\pi}_x) + \lambda d(\sigma_{\hat{\pi}_x}^{x \to y}, \sigma_{\pi_y}^y) \quad (7)$$

where $J(\hat{\pi}_x)$ is the performance of $\hat{\pi}_x$, $d$ is a distance metric between distributions, and $\lambda > 0$ is a Lagrange multiplier. In practice, we found that injectivity of $g$ is unnecessary to enforce in continuous domains. We now present an instantiation of this framework: Generative Adversarial MDP Alignment (GAMA). Recall that we are given an alignment task set $\mathcal{D}_{x,y} = \{(\mathcal{D}_{\mathcal{M}_x, \mathcal{T}_i}, \mathcal{D}_{\mathcal{M}_y, \mathcal{T}_i})\}_{i=1}^N$. In the alignment step, we learn $\pi_{y, \mathcal{T}_i}^*, \forall \mathcal{T}_i$ and parameterized state, action maps $f_{\theta_f} : \mathcal{S}_x \to \mathcal{S}_y, g_{\theta_g} : \mathcal{A}_y \to \mathcal{A}_x$ that compose $\hat{\pi}_{x, \mathcal{T}_i} = g_{\theta_g} \circ \pi_{y, \mathcal{T}_i}^* \circ f_{\theta_f}$. To match $\sigma_{\hat{\pi}_x}^{x \to y}, \sigma_{\pi_y}^y$, we employ adversarial training (Goodfellow

et al., 2014) in which separate discriminators $D_{\theta_D^i}$ per task are trained to distinguish between "real" transitions $(s_y, a_y, s_y') \sim \pi_{y, \mathcal{T}_i}^*$ and "fake" transitions $(\hat{s}_y, \hat{a}_y, \hat{s}_y') \sim \hat{\pi}_{x, \mathcal{T}_i}$, where $\hat{s}_y = f_{\theta_f}(s_x)$, $\hat{a}_y = \pi_y(\hat{s}_y)$, $\hat{s}_y' = f_{\theta_f}(P_{\theta_P}^x(s_x, g(\hat{a}_y)))$, and $P_{\theta_P}^x$ is a fitted model of the $x$ domain dynamics. (see Figure 1(b)) The generator, consisting of $f_{\theta_f}, g_{\theta_g}$, is trained to fool the discriminator while maximizing policy performance. The distribution matching gradients are back propagated through the learned dynamics, $\pi_{y, \mathcal{T}_i}^*$ is learned by Imitation Learing (IL) on $\mathcal{D}_{\mathcal{M}_{y, \mathcal{T}_i}}$, and the policy performance objective on $\hat{\pi}_{x, \mathcal{T}_i}$ is achieved by IL on $\mathcal{D}_{\mathcal{M}_{x, \mathcal{T}_i}}$. In this work we use behavioral cloning (Pomerleau, 1991) for IL. We thus seek to find a saddle point $\{f, g\} \cup \{D_{\theta_D^i}\}_{i=1}^N$ of the following objective:

$$\min_{f, g} \max_{\{D_{\theta_D^i}\}_{i=1}^N} \sum_{i=1}^N \left( \mathbb{E}_{s_x \sim \pi_{x, \mathcal{T}_i}^*} [D_{KL}(\pi_{x, \mathcal{T}_i}^*(\cdot|s_x) || \hat{\pi}_{x, \mathcal{T}_i}(\cdot|s_x))] \right.$$
$$\left. + \lambda (\mathbb{E}_{\pi_{y, \mathcal{T}_i}^*}[\log D_{\theta_D^i}(s_y, a_y, s_y')] + \mathbb{E}_{\pi_{x, \mathcal{T}_i}^*}[\log(1 - D_{\theta_D^i}(\hat{s}_y, \hat{a}_y, \hat{s}_y'))]) \right) \tag{8}$$

where $D_{KL}$ is the KL-divergence. We provide the full execution flow of GAMA in Algorithm 1 In the adaptation step, we are given expert demonstrations $\mathcal{D}_{\mathcal{M}_{y, \mathcal{T}}}$ of a new target task $\mathcal{T}$, from which we fit an expert domain policy $\pi_{y, \mathcal{T}}$ which are composed with the learned alignments to construct an adapted self policy $\hat{\pi}_{x, \mathcal{T}} = g_{\theta_g} \circ \pi_{y, \mathcal{T}} \circ f_{\theta_f}$. We also experiment with a demonstration adaptation method which additionally trains an inverse state map $f^{-1} : \mathcal{S}_y \rightarrow \mathcal{S}_x$, adapts demonstrations $\mathcal{D}_{\mathcal{M}_{y, \mathcal{T}}}$ into the self domain via $f^{-1}, g$, and applies behavioral cloning on the adapted demonstrations. (see Figure 1(c)) Notably, our entire procedure does not require paired, aligned demonstrations nor an RL step.

---

**Algorithm 1:** Generative Adversarial MDP Alignment (GAMA)

1 **input**: Alignment task set $\mathcal{D}_{x,y} = \{(\mathcal{D}_{\mathcal{M}_{x, \mathcal{T}_i}}, \mathcal{D}_{\mathcal{M}_{y, \mathcal{T}_i}})\}_{i=1}^N$ of unpaired trajectories, fitted $\pi_{y, \mathcal{T}_i}^*$

2 **while** not done **do**:

3   **for** $i = 1, ..., N$ **do**:

4     Sample $(s_x, a_x, s_x') \sim \mathcal{D}_{\mathcal{M}_{x, \mathcal{T}_i}}, (s_y, a_y, s_y') \sim \mathcal{D}_{\mathcal{M}_{y, \mathcal{T}_i}}$ and store in buffer $\mathcal{B}_x^i, \mathcal{B}_y^i$

5     **for** $j = 1, ..., M$ **do**:

6       Sample mini-batch $j$ from $\mathcal{B}_x^i, \mathcal{B}_y^i$

7       Update dynamics model with: $-\hat{\mathbb{E}}_{\pi_{x, \mathcal{T}_i}^*}[\nabla_{\theta_P}(P_{\theta_P}^x(s_x, a_x) - s_x')^2]$

8       Update discriminator: $\hat{\mathbb{E}}_{\pi_{y, \mathcal{T}_i}^*}[\nabla_{\theta_D^i} \log D_{\theta_D^i}(s_y, a_y, s_y')] + \hat{\mathbb{E}}_{\pi_{x, \mathcal{T}_i}^*}[\nabla_{\theta_D^i} \log(1 - D_{\theta_D^i}(\hat{s}_y, \hat{a}_y, \hat{s}_y'))]$

9       Update alignments $(f_{\theta_f}, g_{\theta_g})$ with gradients:

$$-\hat{\mathbb{E}}_{\pi_{x, \mathcal{T}_i}^*}[\nabla_{\theta_f} \log D_{\theta_D}(\hat{s}_y, \hat{a}_y, \hat{s}_y')] + \hat{\mathbb{E}}_{\pi_{x, \mathcal{T}_i}^*}[\nabla_{\theta_f}(\hat{\pi}_{x, \mathcal{T}_i}(s_x) - a_x)^2]$$
$$-\hat{\mathbb{E}}_{\pi_{x, \mathcal{T}_i}^*}[\nabla_{\theta_g} \log D_{\theta_D}(\hat{s}_y, \hat{a}_y, \hat{s}_y')] + \hat{\mathbb{E}}_{\pi_{x, \mathcal{T}_i}^*}[\nabla_{\theta_g}(\hat{\pi}_{x, \mathcal{T}_i}(s_x) - a_x)^2]$$

---

**Related Works**: Closely related to CDIL, the field of cross domain transfer learning in the context of RL has explored approaches to use state maps to exploit cross domain demonstrations in a pretraining procedure for a new target task for which self domain reward function is available. Canonical Correlation Analysis (CCA) (Hotelling, 1936) finds invertible projections into a basis in which data from different domains are maximally correlated. These projections can then be composed to obtain a direct correspondence map between states. Ammar et al. (2015); Joshi & Chowdhary (2018) have utilized an unsupervised manifold alignment (UMA) algorithm which finds a linear map between states with similar local geometric properties. UMA assumes the existence of hand crafted features along with a distance metric between them. This family of work commonly uses a linear statemap to define a time-step wise transfer reward and executes an RL step on the new task. Similar to our work, these works use an alignment task set of unpaired, unaligned trajectories to compute the state map. Unlike these works, we learn maps that preserve MDP structure, use deep neural network state, action maps, and achieve zero-shot transfer to the new task without an RL step. More recent work in transfer learning across embodiment (Gupta et al., 2017) and viewpoint (Liu et al., 2018; Sermanet et al., 2018) mismatch obtain state correspondences from an alignment task set comprising paired, time-aligned demonstrations and use them to learn a state map or a state encoder to a domain invariant feature space. In contrast to this family of prior work, our approach learns both state, action maps from *unpaired, unaligned* demonstrations. Also, we remove the need for additional environment interactions and an expensive RL procedure on the target task by leveraging the action map for zero-shot imitation. Stadie et al. (2017) have shown promise in using domain confusion loss and generative adversarial imitation learning (Ho & Ermon, 2016) for learning across small viewpoint mismatch without an alignment task set, but fails in dealing with large viewpoint differences. Unlike Stadie et al. (2017), we leverage the alignment task set to succeed in imitating across

larger viewpoint mismatch and do not require an RL procedure. MDP homomorphisms (Ravindran & Barto, 2002) have been explored with the aim of compressing state, action spaces to facilitate planning. In similar vein, related works have proposed MDP similarity metrics based on bisimulation methods (Ferns et al., 2004) and Boltzmann machine reconstruction error (Ammar et al., 2014). While conceptually related to our MDP alignability theory, these works have not proposed scalable procedures to discover the homomorphisms and have not drawn connections to cross domain learning.

# 6 EXPERIMENTS

Ours experiments were designed to answer the following questions: (1). Can GAMA uncover MDP reductions? (2). Can the learned alignments $(f_{\theta_f}, g_{\theta_g})$ be leveraged to succeed at CDIL? Note that we include experiments with MDP pairs that are *not perfectly alignable*, yet intuitively share structure, to show general applicability of GAMA for CDIL. We propose three metrics to evaluate the effectiveness of GAMA. First, *alignment complexity* which is the number of MDP pairs, i.e number of tasks, in the alignment task set needed to learn alignments that enable zero-shot imitation, given ample cross domain demonstrations for the target tasks. Second, *adaptation complexity* which is the amount of cross domain demonstrations for the target tasks needed to successfully imitate tasks in the self domain without querying the target task reward function, given a sufficiently large alignment task set. Finally, *transferability*, which is the environment sample complexity on the target task when using the alignment procedure as weight initialization then running RL with the target task reward function. While we aim to learn optimal self policies without querying the self domain reward function, this metric measures the usefulness of the alignment step even when MDP pairs in the alignment task set are not in the equivalence class of the target MDP pair. We study two ablations of GAMA and compare against the following baselines:

**GAMA - Policy Adapt (GAMA-PA)**: learns alignments by Algorithm 1, fits an expert policy $\pi_{y,\mathcal{T}}$ to $\mathcal{D}_{\mathcal{M}_{y,\mathcal{T}}}$ for a new target task $\mathcal{T}$ and zero-shot adapts $\pi_{y,\mathcal{T}}$ to the self domain via $\hat{\pi}_{x,\mathcal{T}} = g_{\theta_g} \circ \pi_{y,\mathcal{T}} \circ f_{\theta_f}$.

**GAMA - Demonstration Adapt (GAMA-DA)**: trains $f^{-1}$ in addition to Algorithm 1, adapts $\mathcal{D}_{\mathcal{M}_{y,\mathcal{T}}}$ into the self domain via $(f^{-1}, g)$, and fits a self domain policy on the adapted demonstrations.

**Self Demonstrations (Self-Demo)**: We behavioral clone on self domain demonstrations for the target task. This baseline provides an "upper bound" on the adapation complexity of CDIL.

**Canonical Correlation Analysis (Hotelling, 1936) (CCA)**: finds invertible matrices $C_x, C_y$ to a basis where domain data are maximally correlated from unpaired, unaligned demonstrations.

**Unsupervised Manifold Alignment (Ammar et al., 2015) (UMA)**: finds a map between states that have similar local geometries from unpaired, unaligned demonstrations.

**Invariant Features (Gupta et al., 2017) (IF)**: finds invertible projections onto a feature space given state pairings. Dynamic Time Warping (Muller, 2007) is used to obtain the pairings.

**Imitation from Observation (Liu et al., 2018) (IfO)**: learns a statemap conditioned on a cross domain observation given state pairings. Dynamic Time Warping (Muller, 2007) is used to obtain the pairings.

**Third Person Imitation Learning (Stadie et al., 2017) (TPIL)**: simultaneously learns a domain agnostic feature space and matches distributions in the feature space via GAIL (Ho & Ermon, 2016).

We experiment with environments which are extensions of OpenAI Gym (Brockman et al., 2016). pen, cart, reacher2, reacher3, reach2-tp, snake3, and snake4 denotes the pendulum, cartpole, 2-link reacher, 3-link reacher, third person 2-link reacher, 3-link snake, and 4-link snake environments, respectively. (self domain) $\leftrightarrow$ (expert domain) specify an MDP pair in the alignment task set. Model architectures and environment details are further described in the Supp. Materials, section B, C, D.

## 6.1 MDP ALIGNMENT EVALUATION

Figure 4 visualizes the learned state map $f_{\theta_f}$ for several MDP pairs. The pen $\leftrightarrow$ pen alignment task (Figure 4, Top Left) and reach$\leftrightarrow$reach-tp task task exemplify scenarios where two MDPs are permutations of each other. Similarly, the pen $\leftrightarrow$ cart alignment task (Figure 4, Top Right) has a reduction that maps the pendulum's angle and angular velocity to those of the pole, as the cart's position and velocity are redundant state dimensions once an optimal policy has been learned. Table 1 presents quantitative evaluations of these simple alignment maps. For pen$\leftrightarrow$pen and reach2$\leftrightarrow$reach2-tp we record the average L2 loss between the learned statemap's outputs and the ground truth permutation map's outputs. As for pen$\leftrightarrow$cart, we do

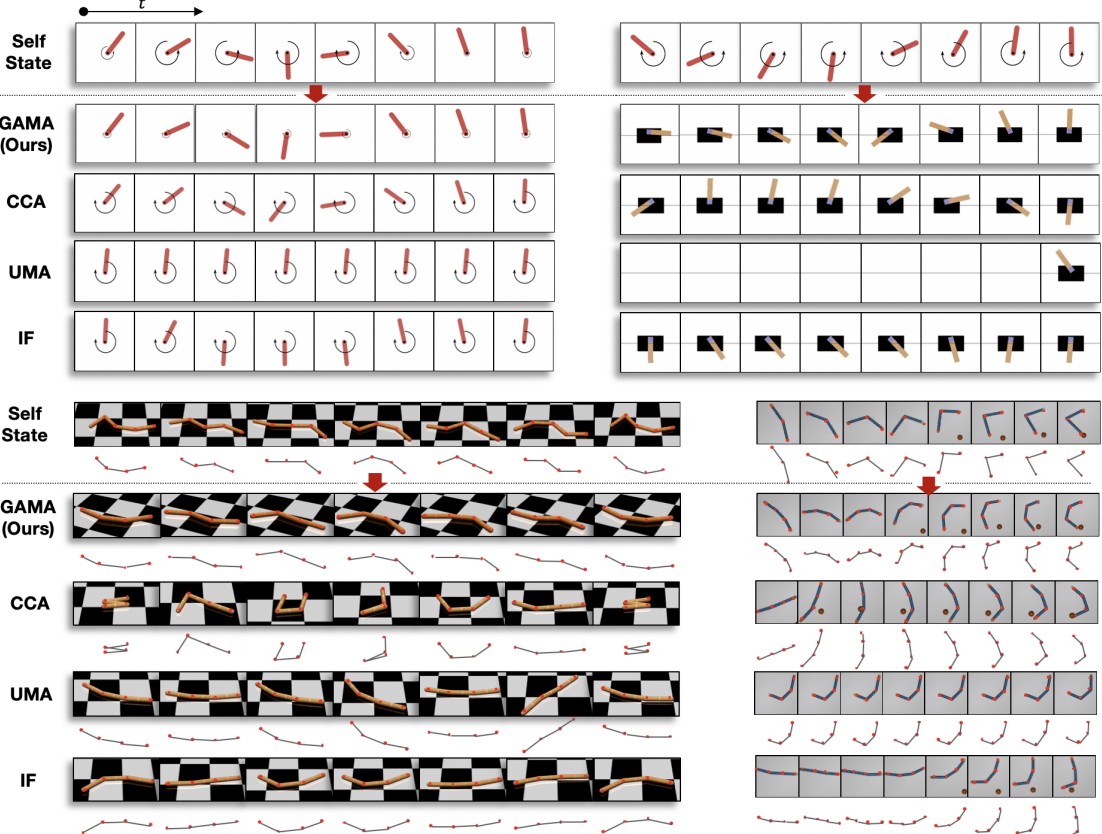

Figure 4: Visualization of the learned state maps for pen↔pen (Top Left), pen↔cart (Top Right), snake4↔snake3 (Bottom Left), reach2↔reach3 (Bottom Right). GAMA is able to recover MDP reductions (Top Left/Right) and finds interpretable correspondences between domains that are not perfectly alignable, yet intuitively share structure (Bottom Left/Right). Baselines fail in most cases

Table 1: Quantitative evaluation of learned state maps. GAMA reliably finds MDP permutations while baselines incur $10\times$ larger deviation loss from the ground truth permutation map. Error bars/regions show the standard deviation over 5 runs.

| | GAMA (ours) | CCA | UMA | IF | IfO | Random |
|---|---|---|---|---|---|---|
| pen ↔ pen | $\mathbf{0.057 \pm 0.017}$ | $0.72 \pm 0.25$ | >100 | $2.50 \pm 1.08$ | $2.24 \pm 0.82$ | >100 |
| pen ↔ cart | $\mathbf{0.178 \pm 0.051}$ | $3.92 \pm 3.77$ | >100 | $1.62 \pm 0.52$ | $3.31 \pm 1.2$ | >100 |
| reach2↔reach2-tp | $\mathbf{0.092 \pm 0.043}$ | $10.14 \pm 5.31$ | >100 | $12.41 \pm 3.12$ | $5.12 \pm 2.41$ | >100 |

the same on the dimensions that correspond to the angle and angular velocity of the pole. We see from both Figure 4 and Table 1 that GAMA is able to learn simple reductions while baselines fail to do so. The key reason behind this performance gap is that most baselines (Gupta et al., 2017; Liu et al., 2018) obtain state maps from time-aligned demonstration data. However, the considered alignment task set contains unaligned demonstrations with diverse starting states, up to 2x differences in demonstration lengths, and varying task execution rates. We see that GAMA also outperforms baselines that learn from unaligned demonstrations (Hotelling, 1936; Ammar et al., 2015) by learning maps that preserve MDP structure with more flexible neural network function approximators. For pen↔cart, UMA learns a statemap that outputs out-of-bounds coordinates mainly because the pendulum demonstrations are concentrated around the pole upright state. The optimal UMA embedding matrix in this case is a zero matrix. Then the UMA state map matrix norm is proportional to the inverse embedding matrix norm which is very large. For snake4 ↔ snake3 and reach2 ↔ reach3, the MDPs may not be perfectly alignable, yet intuitively share structure. From Figure 4 (Bottom Left) we see that GAMA identically matches two adjacent joint angles of snake4 to the two joint angles of snake3 and the periodicity of the snake's wiggle is preserved. On reacher2↔reacher3, we find that the central pivot angles are matched and further find correspondences between states that have similar extents of contraction.

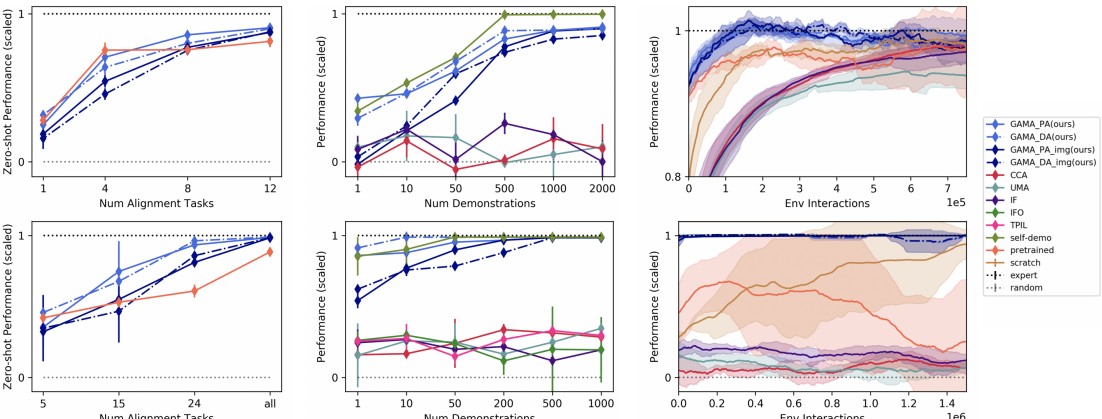

Figure 5: CDIL performance. Alignment complexity (Left), Adaptation complexity (Middle), and transferability (Right) for W2C/R2W on the top/bottom rows, respectively. GAMA outperforms baselines in all metrics. Notably, adaptation complexity of GAMA is close to that of the self-demo baseline. Error bars/regions show the standard deviation over 5 runs.

## 6.2 CDIL PERFORMANCE

**Wall2Corner (W2C)**: The self domain is reacher2 and the expert domain is reacher3. We use the robot's internal state, action representation. The alignment tasks are reaching for 12 goals near the room wall centers and the target tasks are reaching for 12 new goals at the room corners, maximally away from the wall goals. The significant difference between training and test goals makes generalization challenging.

**Reach2Write (R2W)**: The self domain is reacher2 and the expert domain is reacher2-tp that has a "third person" state space with a $180°$ camera angle offset. We use the robot's internal state, action representation. The alignment tasks are reaching for goals and the transfer task is writing letters as fast as possible. The transfer task differs from the alignment tasks in two key aspects: the end effector must draw a straight line from a letter's vertex to vertex and not slow down at the vertices in order to trace the letters fast.

Alignment complexity is shown in Figure 5 (Left). GAMA is able to learn alignments that enable zero-shot imitation on the target task, showing clear gains over a simple pretraining procedure on the self domain MDPs in the alignment task set. Other baselines require an additional RL step and cannot achieve zero-shot imitation. Figure 5 (Middle) shows the adaptation complexity. Notably, GAMA-DA (blue, dashed) produces adapted demonstrations of similar usefulness as self demonstrations (olive green). Other baselines fail to learn useful alignments from unpaired, unaligned demonstrations and as a result fails at CDIL. Finally, Figure 5 (Right) shows that the alignment step is useful as weight initialization to accelerate learning of the target task. GAMA (blue) attains optimal performance around $7\times$ faster than all baselines in the W2C experiment, while immediately attaining optimal performance on the R2W task. Baselines fail to learn the writing task as an inaccurate proxy reward function harms performance.

## 6.3 CDIL WITH VISUAL INPUTS

The non-visual environment experiments in the previous section demonstrate the limitations of the time-alignment assumptions made in prior work without confounding variables such as the difficulty optimization in high-dimensional space. In this section, we also demonstrate that GAMA *scales to higher dimensional, visual environments* with $64 \times 64 \times 3$ image inputs on the W2C and R2W experiments. Specifically, we train a deep spatial autoencoder on the alignment task set to learn an encoder with the architecture from Levine et al. (2016), then apply GAMA on the (learned) latent space. Comparing the dark blue (image input) and light blue curves (internal state input) in Figure 5, we see that the adaptation complexity and alignment complexity of GAMA-DA-img, GAMA-PA-img are both similar to that of GAMA-DA, GAMA-PA and better than baselines trained with the robot's internal representation.

## 7 DISCUSSION AND FUTURE WORK

We've formalized Cross Domain Imitation Learning which encompasses prior work in transfer learning across embodiment (Gupta et al., 2017) and viewpoint differences (Stadie et al., 2017; Liu et al., 2018) along with a practical algorithm that can be applied to both scenarios. We now point out directions future

work. Our MDP alignability theory is a first step towards formalizing possible shared structures that enable cross domain imitation. While we've shown that GAMA empirically works well even when MDPs are not perfectly alignable, upcoming works may explore relaxing the conditions for MDP alignability to develop a theory that covers an even wider range of real world MDPs. Future works may also try applying GAMA in the imitation from observations scenario, i.e actions are not available, by aligning observations with GAMA and applying methods from Sermanet et al. (2018); Liu et al. (2018). Finally, we hope to see future works develop principled ways design a minimal alignment task set, which is analogous to designing a minimal training set for supervised learning.

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

## Cross Domain Imitation Learning - Supplementary Materials

## A    HIGH-LEVEL COMPARISON TO BASELINES

Table 2: Comparison of baselines by attributes demonstrated in the paper. The "No Act" column denotes whether or not the demonstrations need to contain actions.

| | Unpair. Unalign. Data | Zeroshot Imit. | Embod. Mismatch | Viewpoint Mismatch | Single-domain Demo. | No Act. |
|---|---|---|---|---|---|---|
| TPIL (Stadie et al., 2017) | ✓ | ✗ | ✗ | ✓ | ✓ | ✗ |
| IF (Gupta et al., 2017) | ✗ | ✗ | ✓ | ✗ | ✗ | ✗ |
| IfO (Liu et al., 2018) | ✗ | ✗ | ✗ | ✓ | ✗ | ✓ |
| TCN (Sermanet et al., 2018) | ✗ | ✗ | ✗ | ✓ | ✓ | ✓ |
| **GAMA (ours)** | ✓ | ✓ | ✓ | ✓ | ✗ | ✗ |

We note that methods such as IF has potential to be applied to the viewpoint mismatch problem and IfO, TCN have the potential to be applied to the embodiment mismatch problem, albeit they were not shown in the paper. TCN has shown interesting mappings between humans and robots can be learned. However they haven't shown that robots can use these mappings to learn from human demonstrations. Below we summarize the key differences between GAMA and the main baselines.

1. We propose an *unsupervised MDP alignment* algorithm (GAMA) capable of learning state correspondences from *unpaired, unaligned demonstrations* while Gupta et al. (2017); Liu et al. (2018); Sermanet et al. (2018) obtain these correspondences from paired, time-aligned trajectories. Our demonstrations have varying length (up to 2x difference) and diverse starting positions. Since good observation correspondences are prerequisites to the success of Gupta et al. (2017); Liu et al. (2018); Sermanet et al. (2018), our work provide the missing ingredient. Future work could try learning alignments with GAMA, then apply methods from Gupta et al. (2017); Liu et al. (2018); Sermanet et al. (2018) to perform CDIL when action information is unavailable from demonstrations.

2. *We remove the need for an expensive RL procedure on a new target task*, by leveraging action information for zero-shot imitation. By learning a composite self policy with both state and action maps, we obtain a near-optimal self policy on new tasks without any environment interactions while prior approaches (Gupta et al., 2017; Liu et al., 2018; Sermanet et al., 2018) require an additional RL step that involves self domain environment interactions.

3. *We use a single algorithm to address both the viewpoint and embodiment mismatch* which have previously been dealt with different solutions.

## B    GAMA MODEL ARCHITECTURE

We now describe the model architecture. The state, action map $f_{\theta_f}, g_{\theta_g}$, inverse state map $f_{\theta_{f^{-1}}}^{-1}$, transition function $P_{\theta_P}^x$, and discriminators $\{D_{\theta_D^i}\}_{i=1}^N$ are neural networks with hidden layers of size $(200, 200)$. The fitted policies $\{\pi_{y,\mathcal{T}_i}\}_{i=1}^N$ for GAMA-PA and $\pi_{x,\mathcal{T}}$ for GAMA-DA all have hidden layers of size $(300, 200)$. All models are trained with Adam optimizers (Kingma & Ba, 2014) using decay rates $\beta_1 = 0.9, \beta_2 = 0.999$. For the spatial autoencoders used in GAMA-PA-img and GAMA-DA-img, we use the same architecture as in Finn et al. (2015) We use a learning rate of $1e$-4 for the alignment maps and $1e$-5 for all other components. These parameters are fixed across all experiments.

## C    BASELINE IMPLEMENTATION DETAILS

In this section we describe our implementation details of the baselines.

**Obtaining State Correspondences** We use 5000 sampled trajectories in both expert and self domains to learn the state map for IF and CCA. For UMA, we use 20 sampled trajectories to learn that in pendulum and cartpole environment and 50 trajectories in reacher, reacher-tp environment (much beyond these numbers UMA is computationally intractable). For IF and IfO, we use Dynamic Time Warping (DTW) (Muller, 2007) to obtain state correspondences. For IF, DTW uses the (learned) feature space as a metric space to estimate the domain correspondences. For IfO, DTW is applied on the state space. We follow the implementation procedure in Gupta et al. (2017).

To visualize and quantitatively evaluate the statemaps learned in prior work, we compose the encoder and decoder for IF and use the Moore-Penrose pseudo inverse of the embedding matrix for UMA and CCA.

**Transfer Learning** In the transfer learning phase for CCA, UMA, IF, and IfO they define a proxy reward function on the target task by using the state correspondence map.

$$r_{\text{proxy}}(s_x^{(t)}) = \frac{1}{|\mathrm{T}|} \sum_{\tau \in \mathrm{T}} \|f(s_{y,\tau}^{(t)}) - g(s_x^{(t)})\|_2^2$$

, where $s_x^{(t)}$ is a self domain state at time $t$, T is the collection of expert demonstrations, and $s_{y,\tau}^{(t)}$ is the expert domain state at time $t$ in trajectory $\tau$. IfO additionally defines a penalty reward for deviating from states encountered during training. We refer readers to their paper (Liu et al., 2018) for further details. The transferability results of Figure 5 (Right) show the learning curve for training on the ground truth reward for the target task where the policy is pretrained with a training procedure on the proxy reward. All RL steps are performed using the Deep Deterministic Policy Gradient (DDPG) (Lillicrap et al., 2015) algorithm.

**Architecture** For UMA and CCA, the embedding dimension is the minimum state dimension between the expert and self domains. For UMA, we use one state sample every 5 timesteps to reduce the computational time, and we match the pairwise distance matrix of 3-nearest neighbors.

For IF, we use 2 hidden layer with 64 hidden units each and leaky ReLU non-linearities to parameterize embedding function and decoders, the dimension of common feature space is set to be 64. The optimizer are same with respect to our models and the learning rate is $1e$-3.

For IfO, we use the same architecture as the statemap in GAMA for their observation conditioned statemap.

For TPIL, we use 3 hidden layer with 128 hidden units each and ReLU non-linearities to parameterize the feature extractor, classifier and domain classifier. We use Adam Optimizer with default decay rates and learning rate $1e$-3 to train the discriminator and use same optimizer and learning rate with respect to our model to train the policy.

## D    ENVIRONMENTS

We use the 'Pendulum-v0', 'Cartpole-v0' environments for the pendulum and cartpole tasks which have state space $(w, \dot{w})$ and $(w, \dot{w}, x, \dot{x})$, respectively, where $w$ is the angle of the pendulum/pole and $x$ is the position of the cart. The action spaces are $(F_w)$ and $(F_x)$ where $F_w$ is the torque applied to the pendulum's pivot and $F_x$ is the x-direction force applied to the cart. For snake3, snake4 we use an extension Wang et al. (2018) of the 'Swimmer-v0' environment from Gym Brockman et al. (2016). A $K$ link snake has a state representation $(w_1, ..., w_K, \dot{w}_1, ..., \dot{w}_K)$ where $w_k$ is the angle of the $k^{th}$ snake joint. The action vector has the form $(F_{w_1}, ...F_{w_K})$ where $F_{w_k}$ is the torque applied to joint $k$. All reacher environments were extended from the 'Reacher-v0' gym environment. A $k$ link reacher has a state vector of the form $(w_1, ..., w_K, \dot{w}_1, ..., \dot{w}_K, x_g, y_g)$ where $w_k$ is the angle of the $k^{th}$ reacher joint and $(x_g, y_g)$ is the position of the goal. Note the key difference with the original Reacher-v0 environment is that the difference vector between the end effector and the goal coordinate was removed from the state to make the task more challenging. The action vector has the form $(F_{w_1}, ...F_{w_K})$ where $F_{w_k}$ is the torque applied to joint $k$. The third person reacher environment (reach-tp) has an angle off set of $\pi$ such that $(w_1 + \pi, ..., w_K + \pi, \dot{w}_1, ..., \dot{w}_K, x_g, y_g)$ corresponds to the same internal robot state as $(w_1, ..., w_K, \dot{w}_1, ..., \dot{w}_K, x_g, y_g)$ in the original reacher domain. Note that the goal coordinates $(x_g, y_g)$ have not changed but the joint angles have changed. Thus a proper alignment between the original domain and the third person domain should add an offset of $\pi$ to the original state. For the writing task, the vertices of the letters to write are specified sequentially with the goal coordinates. Once the first vertex is reached, the goal coordinates are updated to be the next vertex coordinates. The reward function is defind as follows:

$$R_{write}(s) = \begin{cases} 100 & \text{if state } s \text{ corresponds to reaching a vertex} \\ -1 & \text{else} \end{cases}$$

Thus the agent must perform a sequential reaching task and accomplish it as fast as possible. The key difference with a normal reaching task is that the reacher must not slow down at each vertex and plan it's path accordingly in order to minimize drastic direction changes. Further more the reward is significantly more sparse than the original reacher reward which gets reward inversely proportional to the distance between the end effector and the goal.

## E   VIDEOS OF ALIGNMENT MAPS

We will provide real-time visualizations of the learned alignments in the final version of the submission.

## F   PROOFS

**Definition 1.** *An optimality function $O_{\mathcal{M}_x} : \mathcal{S}_x \times \mathcal{A}_x \rightarrow \{0,1\}$ for an MDP $\mathcal{M}_x$ satisfies: $O_{\mathcal{M}_x}(s_x, a_x) = 1$ if $\exists \pi_x^* \in \Pi_{\mathcal{M}_x}^*$ such that $(s_x, a_x) \in \mathrm{supp}(q_{\pi_x^*})$ and $O_{\mathcal{M}_x}(s_x, a_x) = 0$ otherwise.*

**Definition 6.** *An optimal policy $\pi_x$ is **covering** if $O_{\mathcal{M}_x}(s_x, a_x) = 1 \Rightarrow a_x \in supp(\pi_x(\cdot|s_x))$.*

We first restate the definition of MDP reductions:

**Definition 2.** *An **MDP reduction** from $\mathcal{M}_x = (\mathcal{S}_x, \mathcal{A}_x, P_x, \eta_x, R_x)$ to $\mathcal{M}_y = (\mathcal{S}_y, \mathcal{A}_y, P_y, \eta_y, R_y)$ is a tuple $r = (\phi, \psi)$ where $\phi : \mathcal{S}_x \rightarrow \mathcal{S}_y, \psi : \mathcal{A}_x \rightarrow \mathcal{A}_y$ are maps that preserve:*

*1. (optimal policy) $\forall s_x \in \mathcal{S}_x, a_x \in \mathcal{A}_x, s_y \in \mathcal{S}_y, a_y \in \mathcal{A}_y$,*

$$O_{\mathcal{M}_y}(\phi(s_x), \psi(a_x)) = 1 \quad \Rightarrow \quad O_{\mathcal{M}_x}(s_x, a_x) = 1 \tag{1}$$

$$O_{\mathcal{M}_y}(s_y, a_y) = 1 \quad \Rightarrow \quad \phi^{-1}(s_y) \neq \emptyset, \psi^{-1}(a_y) \neq \emptyset \tag{2}$$

*2. (dynamics) $\forall s_y, s_y' \in \mathcal{S}_y, a_y \in \mathcal{A}_y$ where $O_{\mathcal{M}_y}(s_y, a_y) = 1$,*

$$P_y(s_y, a_y) = \phi(P_x(s_x, a_x)) \quad \forall s_x \in \phi^{-1}(s_y), a_x \in \psi^{-1}(a_y) \tag{3}$$

*where we define $\phi^{-1}(s_y) = \{s_x | \phi(s_x) = s_y\}$, $\psi^{-1}(a_y) = \{a_x | \psi(a_x) = a_y\}$. Furthermore, $r$ is an **MDP permutation** if and only if $\phi, \psi$ are bijective maps.*

**Definition 7.** *MDP $\mathcal{M}_x$ is **unichain**, if all policies induce irreducible Markov Chains and all stochastic optimal policies induce ergodic, i.e irreducible and aperiodic, Markov Chains.*

**Assumption 1.** *All considered MDPs are unichain with discrete state, action spaces and deterministic dynamics i.e. $P : \mathcal{S} \times \mathcal{A} \rightarrow \mathcal{S}$. Furthermore, there exists dummy state, actions $s^d, a^d$ where $O_{\mathcal{M}}(s, a^d) = 0 \ \forall s \in \mathcal{S}$ and $O_{\mathcal{M}}(s^d, a) = 0 \ \forall a \in \mathcal{A}$*

As stated in Assumption 1, we specifically consider MDPs that are unichain with deterministic dynamics. We note this is a reasonable assumption since physics is largely deterministic and many control behaviors, such as walking, are described by unichains. We now formalize a notion of MDP alignability starting from defining a class of structure preserving maps between MDPs. We first prove some lemmas that will assist in proving the main Theorem. Recall that $f : \mathcal{S}_x \rightarrow \mathcal{S}_y, g : \mathcal{A}_y \rightarrow \mathcal{A}_x$ and $\hat{\pi}_x = g \circ \pi_y \circ f$

**Lemma 1.** *Let $\mathcal{M}_x, \mathcal{M}_y$ be MDPs satisfying Assumption 1 (see Supp. Materials), $\mathcal{M}_x \geq_{\phi, \psi} \mathcal{M}_y$, and $\pi_y$ be optimal in $\mathcal{M}_y$. $\forall g : \mathcal{A}_y \rightarrow \mathcal{A}_x$ s.t $\psi \circ g(a_y) = a_y \ \forall a_y \in \{a_y | \exists s_y \in \mathcal{S}_y \ s.t \ O_{\mathcal{M}_y}(s_y, a_y) = 1\}$, it holds that $\hat{\pi}_x = g \circ \pi_y \circ \phi$ is optimal in $\mathcal{M}_x$.*

*Proof.* Without loss of generality, consider an arbitrarily chosen sample $a_x = g(a_y), a_y \sim \pi_y(\cdot|\phi(s_x))$ for any $s_x \in \mathcal{S}_x$. We first see that:

$$O_{\mathcal{M}_y}(\phi(s_x), \psi(a_x)) = O_{\mathcal{M}_y}(\phi(s_x), \psi(g(a_y))) = O_{\mathcal{M}_y}(\phi(s_x), a_y) = 1 \tag{9}$$

where the first step substitutes $a_x = g(a_y)$, the second step applies $\psi \circ g(a_y) = a_y$ since $O(\phi(s_x), a_y) = 1$ due to the optimality of $\pi_y$, and the last step follows from Corollary 1. Since $(\phi, \psi)$ is a reduction, we have that $O_{\mathcal{M}_y}(\phi(s_x), \psi(a_x)) = 1 \Rightarrow O_{\mathcal{M}_x}(s_x, a_x) = 1$ by Equation (1). Therefore, $O_{\mathcal{M}_x}(s_x, a_x) = 1 \ \forall s_x \in \mathcal{S}_x, \forall a_x \in \mathrm{supp}(\hat{\pi}_x(\cdot|s_x))$. Then by Lemma 2, $\hat{\pi}_x$ is optimal.   $\square$

**Lemma 2.** *Let MDP $\mathcal{M}_x$ satisfy Assumption 1 and $\pi_x(a_x|s_x)$ be a (stochastic) mixture policy that chooses $a_x$ randomly from $\{a_x | O(s_x, a_x) = 1\}$. Then, $\pi_x$ is optimal. (Ortner, 2005)*

**Corollary 1.** *Let MDP $\mathcal{M}_x$ satisfy Assumption 1 and $\pi_x$ be optimal. Then $O_{\mathcal{M}_x}(s_x, a_x) = 1 \ \forall s_x \in \mathcal{S}_x, a_x \in supp(\pi_x(\cdot|s_x))$*

**Lemma 3.** *Let MDP $\mathcal{M}_x$ satisfy Assumption 1 and $\pi_x$ be a stochastic optimal policy. Then the triplet stationary distribution $\rho_{\pi_x}^x(s_x, a_x, s_x') = \lim_{t \rightarrow \infty} \Pr(s_x^{(t)} = s_x, a_x^{(t)} = a_x, s_x^{(t+1)} = s_x'; \pi_x, P_x, \eta_x)$ exists and is unique.*

*Proof.*

$$\rho_{\pi_x}^x(s_x, a_x, s_x') = \lim_{t \to \infty} \Pr(s_x^{(t)} = s_x, a_x^{(t)} = a_x, s_x^{(t+1)} = s_x'; \pi_x, P_x, \eta_x)$$

$$= \lim_{t \to \infty} \Pr(s_x^{(t)} = s_x; \pi_x, P_x, \eta_x)\pi_x(a_x|s_x)\mathbb{1}(s_x' = P_x(s_x, a_x))$$

$$= \pi_x(a_x|s_x)\mathbb{1}(s_x' = P_x(s_x, a_x)) \lim_{t \to \infty} \Pr(s_x^{(t)} = s_x; \pi_x, P_x, \eta_y)$$

where $\mathbb{1}$ is the indicator function. The limit in the last line is the stationary distribution over states, which exists and is unique since a stochastic optimal policy induces an ergodic Markov Chain over states. $\square$

**Lemma 4.** *If a real sequence $\{a_i\}_{i=1}^{\infty}$ converges to some $a \in \mathbb{R}$, then*

$$\lim_{T \to \infty} \frac{1}{T} \sum_{i=1}^{T} a_i = \lim_{i \to \infty} a_i = a$$

*Proof.* Denote $A_T = \sum_{i=1}^{T} a_i$, and $B_T = T$. We have

$$\lim_{T \to \infty} \frac{A_{T+1} - A_T}{B_{T+1} - B_T} = \lim_{T \to \infty} a_{T+1} = a \tag{10}$$

According to the Stolz–Cesàro theorem,

$$\lim_{T \to \infty} \frac{A_{T+1} - A_T}{B_{T+1} - B_T} = \lim_{T \to \infty} \frac{A_T}{B_T}$$

if the limit on the left hand side exists. Therefore

$$\lim_{T \to \infty} \frac{A_T}{B_T} = \lim_{T \to \infty} \frac{1}{T} \sum_{i=1}^{T} a_i = a \tag{11}$$

which completes the proof. $\square$

Recall that our target distribution $\sigma_{\pi_y}^y$ and proxy distribution $\sigma_{\hat{\pi}_x}^{x \to y}$ were defined as:

$$\sigma_{\pi_y}^y(s_y, a_y, s_y') = \lim_{T \to \infty} \frac{1}{T} \sum_{t=0}^{T-1} \Pr(s_y^{(t)} = s_y, a_y^{(t)} = a_y, s_y^{(t+1)} = s_y'; \pi_y, P_y, \eta_y) \tag{12}$$

$$\sigma_{\hat{\pi}_x}^{x \to y}(s_y, a_y, s_y') = \lim_{T \to \infty} \frac{1}{T} \sum_{t=0}^{T-1} \Pr(\hat{s}_y^{(t)} = s_y, \hat{a}_y^{(t)} = a_y, \hat{s}_y^{(t+1)} = s_y'; \mathcal{P}) \tag{13}$$

We are now ready to prove that our proxy and target limiting distributions exist.

**Lemma 5.** *Let MDP $\mathcal{M}_y$ satisfy Assumption 1 and $\pi_y$ be a stocahstic optimal policy. Then, $\sigma_{\pi_y}^y(s_y, a_y, s_y') = \rho_{\pi_y}^y(s_y, a_y, s_y').$*

*Proof.* Recall that the stationary distribution $\rho_{\pi_y}^y(s_y, a_y, s_y')$ is the following limiting distribution:

$$\rho_{\pi_y}^y(s_y, a_y, s_y') = \lim_{t \to \infty} \Pr(s_y^{(t)} = s_y, a_y^{(t)} = a_y, s_y^{(t+1)} = s_y'; \pi_y, P_y, \eta_y) \tag{14}$$

$\rho_{\pi_y}^y(s_y, a_y, s_y')$ exist for $\mathcal{M}_y$ as shown in Lemma 3. Then,

$$\sigma_{\pi_y}^y(s_y, a_y, s_y') = \lim_{T \to \infty} \frac{1}{T} \sum_{t=0}^{T-1} \Pr(s_y^{(t)} = s_y, a_y^{(t)} = a_y, s_y^{(t+1)} = s_y'; \pi_y, P_y, \eta_y) \tag{15}$$

$$= \lim_{t \to \infty} \Pr(s_y^{(t)} = s_y, a_y^{(t)} = a_y, s_y^{(t+1)} = s_y'; \pi_y, P_y, \eta_y) \tag{16}$$

$$= \rho_{\pi_y}^y(s_y, a_y, s_y') \tag{17}$$

as desired. The second line follows from Lemma 4 and the last line follows from Lemma 3. $\square$

**Lemma 6.** *Let MDP $\mathcal{M}_y$ satisfy Assumption 1 and $\pi_y$ be a stochastic optimal policy. Then,*

$$supp(\sigma_{\pi_y}^y) \subseteq \{(s_y, a_y, s_y')|O_{\mathcal{M}_y}(s_y, a_y) = 1, s_y, s_y' \in \mathcal{S}_y, a_y \in \mathcal{A}_y\}$$

*Proof.* Assume for contradiction that there exists $(s_y, a_y, s'_y) \in \text{supp}(\sigma_{\pi_y}^y)$ but $(s_y, a_y, s'_y) \notin \{(s_y, a_y, s'_y)|O_{\mathcal{M}_y}(s_y, a_y) = 1, s_y, s'_y \in \mathcal{S}_y, a_y \in \mathcal{A}_y\}$. Then $O_{\mathcal{M}_y}(s_y, a_y) = 0$. Since

$$
\begin{aligned}
\sigma_{\pi_y}^y(s_y, a_y, a_y) &= \lim_{t\to\infty} \Pr(s_y^{(t)} = s_y, a_y^{(t)} = a_y, s_y^{(t+1)} = s'_y; \pi_y, P_y, \eta_y) \\
&= \lim_{t\to\infty} \Pr(s_y^{(t)} = s_y) \Pr(a_y^{(t)} = a_y|s_y^{(t)} = s_y) \Pr(s_y^{(t+1)} = s'_y|s_y^{(t)} = s_y, a_y^{(t)} = a_y) \\
&= \lim_{t\to\infty} \Pr(s_y^{(t)} = s_y) \pi_y(a_y|s_y) \Pr(s_y^{(t+1)} = s'_y|s_y^{(t)} = s_y, a_y^{(t)} = a_y) \\
&= \underbrace{\pi_y(a_y|s_y)}_{0} \Pr(s_y^{(t+1)} = s'_y|s_y^{(t)} = s_y, a_y^{(t)} = a_y) \lim_{t\to\infty} \Pr(s_y^{(t)} = s_y) \\
&= 0
\end{aligned}
$$

First line follows from Lemma 5 and terms are taken out of the limit in the second to last line since the stationary distribution over states exist as $\mathcal{M}_y$ is unichain and $\pi_y$ is stochastic optimal. $\pi_y(a_y|s_y) = 0$ since $O_{\mathcal{M}_y}(s_y, a_y) = 0 \Rightarrow \pi_y(a_y|s_y) = 0$ from Corollary 1. Then, we have $\sigma_{\pi_y}^y(s_y, a_y, a_y) = 0$ which contradicts $(s_y, a_y, s'_y) \in \text{supp}(\sigma_{\pi_y}^y)$ concluding the proof. □

**Lemma 7.** *Let MDP $\mathcal{M}_x$ satisfy Assumption 1 and $\hat{\pi}_x = g \circ \pi_y \circ f$ be an stochastic optimal policy in $\mathcal{M}_x$ where $f : \mathcal{S}_x \to \mathcal{S}_y$ is the state map, $g : \mathcal{A}_y \to \mathcal{A}_x$ is injective action map, and $\pi_y$ is a stochastic optimal policy in $\mathcal{M}_y$. Further let $\mathcal{F} : \mathcal{S}_x \times g(\mathcal{A}_y) \times \mathcal{S}_x \to \mathcal{S}_y \times \mathcal{A}_y \times \mathcal{S}_y$ be the map $\mathcal{F}(a, b, c) = (f(a), g^{-1}(b), f(c))$. Then, $\sigma_{\hat{\pi}_x}^{x \to y}(s_y, a_y, s_y') = \mathcal{F}(\rho_{\hat{\pi}_x}^x(s_x, a_x, s_x'))$.*

*Proof.* We first define the triplet random variables $X^{(t)} = (s_x^{(t)}, a_x^{(t)}, s_x^{(t+1)})$ for $t = 0, 1, 2, ...$ where $s_x^{(t)}, a_x^{(t)}, s_x^{(t+1)}$ for $t = 0, 1, 2, ...$ were defined in Definition 5. $\mathcal{F}$ is a function on $supp(\rho_{\hat{\pi}_x}^x) \in \mathcal{S}_x \times g(\mathcal{A}_y) \times \mathcal{S}_x$ and $\mathcal{F}(X^{(t)}) = (\hat{s}_y^{(t)}, \hat{a}_y^{(t)}, \hat{s}_y^{(t+1)})$. Furthermore, since $\mathcal{F}$ is a function defined on a discrete domain and codomain, there always exists a trivial continuous extension of $\mathcal{F}$. We may thus apply the continuous mapping theorem (Billingsley, 1968):

$$X^{(t)} \xrightarrow{d} X \Rightarrow \mathcal{F}(X^{(t)}) \xrightarrow{d} \mathcal{F}(X)$$

Since $\mathcal{M}_x$ is unichain and $\hat{\pi}_x$ is stochastic optimal, the distribution of $X^{(t)}$ converges (in distribution) to $\rho_{\hat{\pi}_x}^x(s_x, a_x, s_x')$ as $t \to \infty$ by Lemma 3. Applying the continuous mapping theorem, it follows that the distribution of $\mathcal{F}(X^{(t)}) = (\hat{s}_y^{(t)}, \hat{a}_y^{(t)}, \hat{s}_y^{(t+1)})$ converges (in distribution) to the pushforward measure $\mathcal{F}(\rho_{\hat{\pi}_x}^x(s_x, a_x, s_x'))$ as $t \to \infty$

Directly applying this result, we obtain:

$$\sigma_{\hat{\pi}_x}^{x \to y}(s_y, a_y, s_y') = \lim_{T \to \infty} \frac{1}{T} \sum_{t=0}^{T-1} \Pr(\hat{s}_y^{(t)} = s_y, \hat{a}_y^{(t)} = a_y, \hat{s}_y^{(t+1)} = s_y'; \mathcal{P}) \tag{18}$$

$$= \lim_{t \to \infty} \Pr(\hat{s}_y^{(t)} = s_y, \hat{a}_y^{(t)} = a_y, \hat{s}_y^{(t+1)} = s_y'; \mathcal{P}) \tag{19}$$

$$= \mathcal{F}(\rho_{\hat{\pi}_x}^x(s_x, a_x, s_x')) \tag{20}$$

as desired. Line $(18) \to (19)$ follows from Lemma 4 and $(19) \to (20)$ follows from the continuous mapping theorem. $\square$

**Lemma 8.** *Let $X, Y$ be countable sets, $\phi : X \to Y$ be a function, and $\mathbb{1}$ be the indicator function. We denote $\phi^{-1}(y) = \{x | \phi(x) = y\}$. Then $\forall x \in X, y \in Y$*

$$\mathbb{1}(y = \phi(x)) = \sum_{z \in \phi^{-1}(y)} \mathbb{1}(x = z)$$

*Proof.* Since both the left and right hand-side of the desired equality only take on values in $\{0, 1\}$, it suffices to show the following statements hold for arbitrarily chosen $x \in X, y \in Y$:

$$\sum_{z \in \phi^{-1}(y)} \mathbb{1}(x = z) = 1 \Rightarrow \mathbb{1}(y = \phi(x)) = 1$$

$$\mathbb{1}(y = \phi(x)) = 1 \Rightarrow \sum_{z \in \phi^{-1}(y)} \mathbb{1}(x = z) = 1$$

For the first direction, we see that if $\sum_{z \in \phi^{-1}(y)} \mathbb{1}(x = z) = 1$, then $x \in \phi^{-1}(y)$, and thus $\phi(x) = y$.

For the second direction if $\mathbb{1}(y = \phi(x)) = 1$, then $x \in \phi^{-1}(y)$. Thus there exists a unique $z$ such that $z = x$ and $z \in \phi^{-1}(y)$. Then, $\sum_{z \in \phi^{-1}(y)} \mathbb{1}(x = z) = 1$ as desired, which concludes the proof. $\square$

Finally, we prove the main theorem. Our objectives are:

1. $\hat{\pi}_x$ is optimal
2. $\sigma_{\hat{\pi}_x}^{x \to y} = \sigma_{\pi_y}^y$
3. $g$ is injective

**Theorem 1.** *Let $\mathcal{M}_x, \mathcal{M}_y$ be MDPs satisfying Assumption 1 (see Supp Materials). If $\mathcal{M}_x \geq \mathcal{M}_y$, then $\exists f : \mathcal{S}_x \to \mathcal{S}_y, g : \mathcal{A}_y \to \mathcal{A}_x$, and an optimal covering policy $\pi_y$ (Supp Materials, Def 6) that satisfy objectives 1, 2. Conversely, if $\exists f : \mathcal{S}_x \to \mathcal{S}_y, g : \mathcal{A}_y \to \mathcal{A}_x$ and an optimal covering policy $\pi_y$ satisfying objectives 1, 2, 3, then $\mathcal{M}_x \geq \mathcal{M}_y$ and $\exists(\phi, \psi) \in \Gamma(\mathcal{M}_x, \mathcal{M}_y)$ s.t $f = \phi$ and $\psi \circ g(a_y) = a_y, \forall a_y \in \mathcal{A}_y$.*

*Proof.* We first show the ($\Rightarrow$) direction. Using any $(\phi, \psi) \in \Gamma(\mathcal{M}_x, \mathcal{M}_y)$ we construct $f$ and $g$ in the following manner: $f(s_x) = \phi(s_x) \ \forall s_x \in \mathcal{S}_x$. $g(a_y)$ maps to an arbitrary chosen element from the set $\psi^{-1}(a_y) = \{a_x | \psi(a_x) = a_y\}$ if $\psi^{-1}(a_y) \neq \emptyset$ and an arbitrarily chosen action $a_x \in \mathcal{A}_x$ otherwise. We see that $\forall a_y \in \mathcal{A}_y$ for which $\exists s_y \in \mathcal{S}_y$ such that $O_{\mathcal{M}_y}(s_y, a_y) = 1$, it holds that $\psi^{-1}(a_y) \neq \emptyset$ by Eq 2. Therefore, $\psi \circ g(a_y) = a_y \ \forall a_y \in \mathcal{A}_y$ for which $\exists s_y$ such that $O_{\mathcal{M}_y}(s_y, a_y) = 1$ since $\psi$ maps all elements in $\psi^{-1}(a_y)$ to $a_y$. For $\pi_y$ we choose any covering optimal policy for $\mathcal{M}_y$. It suffices to show that this choice of $f, g, \pi_y$ satisfies objectives 1, 2.

• Objective 1. $\hat{\pi}_x$ is optimal: follows from Lemma 1.

• Objective 2. $\sigma_{\hat{\pi}_x}^{x \to y} = \sigma_{\pi_y}^y$: Since $f = \phi$ is a reduction, it follows that $\forall s_y \in \mathcal{S}_y, a_y \in \mathcal{A}_y$ such that $O_{\mathcal{M}_y}(s_y, a_y) = 1$, any $s'_y \in \mathcal{S}_y$, and $\forall t = 0, 1, 2, ...$:

$$\Pr(\hat{s}_y^{(t+1)} = s'_y | \hat{s}_y^{(t)} = s_y, \hat{a}_y^{(t)} = a_y)$$

$$= \sum_{s'_x \in \mathcal{S}_x} \Pr(\hat{s}_y^{(t+1)} = s'_y | \hat{s}_x^{(t+1)} = s'_x, \hat{s}_y^{(t)} = s_y, \hat{a}_y^{(t)} = a_y) \Pr(\hat{s}_x^{(t+1)} = s'_x | \hat{s}_y^{(t)} = s_y, \hat{a}_y^{(t)} = a_y)$$

$$= \sum_{s'_x \in \mathcal{S}_x} \Pr(\hat{s}_y^{(t+1)} = s'_y | \hat{s}_x^{(t+1)} = s'_x)$$
$$\sum_{\substack{s_x \in \mathcal{S}_x \\ a_x \in \mathcal{A}_x}} \Pr(\hat{s}_x^{(t+1)} = s'_x | s_x^{(t)} = s_x, a_x^{(t)} = a_x, \hat{s}_y^{(t)} = s_y, \hat{a}_y^{(t)} = a_y) \Pr(s_x^{(t)} = s_x, a_x^{(t)} = a_x | \hat{s}_y^{(t)} = s_y, \hat{a}_y^{(t)} = a_y)$$

$$= \sum_{s'_x \in \mathcal{S}_x} \mathbb{1}(s'_y = \phi(s'_x))$$
$$\sum_{\substack{s_x \in \mathcal{S}_x \\ a_x \in \mathcal{A}_x}} \Pr(\hat{s}_x^{(t+1)} = s'_x | s_x^{(t)} = s_x, a_x^{(t)} = a_x) \Pr(s_x^{(t)} = s_x | a_x^{(t)} = a_x, \hat{s}_y^{(t)} = s_y, \hat{a}_y^{(t)} = a_y) \Pr(a_x^{(t)} = a_x | \hat{s}_y^{(t)} = s_y, \hat{a}_y^{(t)} = a_y)$$

$$= \sum_{s'_x \in \mathcal{S}_x} \mathbb{1}(s'_y = \phi(s'_x)) \sum_{\substack{s_x \in \mathcal{S}_x \\ a_x \in \mathcal{A}_x}} \mathbb{1}(s'_x = P_x(s_x, a_x)) \Pr(s_x^{(t)} = s_x | \hat{s}_y^{(t)} = s_y) \Pr(a_x^{(t)} = a_x | \hat{a}_y^{(t)} = a_y)$$

$$= \sum_{s'_x \in \mathcal{S}_x} \mathbb{1}(s'_y = \phi(s'_x)) \sum_{\substack{s_x \in \mathcal{S}_x \\ a_x \in \mathcal{A}_x}} \mathbb{1}(s'_x = P_x(s_x, a_x)) \frac{\Pr(\hat{s}_y^{(t)} = s_y | s_x^{(t)} = s_x) \Pr(s_x^{(t)} = s_x)}{\sum_{s''_x \in \mathcal{S}_x} \Pr(\hat{s}_y^{(t)} = s_y | s_x^{(t)} = s''_x) \Pr(s_x^{(t)} = s''_x)} \mathbb{1}(a_x = g(a_y))$$

$$= \sum_{s'_x \in \mathcal{S}_x} \mathbb{1}(s'_y = \phi(s'_x)) \sum_{s_x \in \mathcal{S}_x} \mathbb{1}(s'_x = P_x(s_x, g(a_y))) \frac{\mathbb{1}(s_y = \phi(s_x)) \Pr(s_x^{(t)} = s_x)}{\sum_{s''_x \in \mathcal{S}_x} \mathbb{1}(s_y = \phi(s''_x)) \Pr(s_x^{(t)} = s''_x)}$$

$$= \sum_{s'_x \in \phi^{-1}(s'_y)} \sum_{s_x \in \phi^{-1}(s_y)} \mathbb{1}(s'_x = P_x(s_x, g(a_y))) \frac{\Pr(s_x^{(t)} = s_x)}{\sum_{s''_x \in \phi^{-1}(s_y)} \Pr(s_x^{(t)} = s''_x)}$$

$$= \sum_{s_x \in \phi^{-1}(s_y)} \frac{\Pr(s_x^{(t)} = s_x)}{\sum_{s''_x \in \phi^{-1}(s_y)} \Pr(s_x^{(t)} = s''_x)} \sum_{s'_x \in \phi^{-1}(s'_y)} \mathbb{1}(s'_x = P_x(s_x, g(a_y)))$$

$$\overset{\text{Lemma8}}{=} \sum_{s_x \in \phi^{-1}(s_y)} \frac{\Pr(s_x^{(t)} = s_x)}{\sum_{s''_x \in \phi^{-1}(s_y)} \Pr(s_x^{(t)} = s''_x)} \mathbb{1}\left(s'_y = \phi\left(P_x\left(s_x, g(a_y)\right)\right)\right)$$

$$\overset{\text{Eq3}}{=} \sum_{s_x \in \phi^{-1}(s_y)} \frac{\Pr(s_x^{(t)} = s_x)}{\sum_{s''_x \in \phi^{-1}(s_y)} \Pr(s_x^{(t)} = s''_x)} \mathbb{1}(s'_y = P_y(s_y, a_y))$$

$$= \mathbb{1}(s'_y = P_y(s_y, a_y))$$

$$= \Pr(s_y^{(t+1)} = s'_y | s_y^{(t)} = s_y, a_y^{(t)} = a_y) \tag{21}$$

Furthermore, from Definition 5, we have:

$$\Pr(\hat{a}_y^{(t)} = a_y | \hat{s}_y^{(t)} = s_y) = \pi_y(a_y | s_y) = \Pr(a_y^{(t)} = a_y | s_y^{(t)} = s_y) \tag{22}$$

Then, $\forall s_y, s'_y \in \mathcal{S}_y$ and $\forall t = 0, 1, 2, ...$

$$\Pr(\hat{s}_y^{(t+1)} = s_y | \hat{s}_y^{(t)} = s_y) = \sum_{a_y \in \mathcal{A}_y} \Pr(\hat{s}_y^{(t+1)} = s_y | \hat{s}_y^{(t)} = s_y, \hat{a}_y^{(t)} = a_y) \Pr(\hat{a}_y^{(t)} = a_y | \hat{s}_y^{(t)} = s_y)$$

$$= \sum_{a_y \in \mathrm{supp}(\pi_y(\cdot | s_y))} \Pr(s_y^{(t+1)} = s_y | s_y^{(t)} = s_y, a_y^{(t)} = a_y) \pi_y(a_y | s_y)$$

$$= \Pr(s_y^{(t+1)} = s_y | s_y^{(t)} = s_y) \tag{23}$$

we are justified in the substitution for the dynamics in the second line since $O_{\mathcal{M}_y}(s_y, a_y) = 1 \forall s_y \in \mathcal{S}_y, a_y \in \mathrm{supp}(\pi_y(\cdot | s_y))$ by Corollary 1. Since $\mathcal{M}_y$ is unichain and $\pi_y$ is a stochastic optimal policy, the stationary distribution $\lim_{t \to \infty} \Pr(s_y^{(t)} = s_y)$ is invariant to the initial state distribution $\eta_y$ and only depends on the state transition dynamics $\Pr(s_y^{(t+1)} = s'_y | s_y^{(t)} = s_y)$. Equivalently any stochastic process with the same state transition dynamics will converge to the same stationary distribution regardless of the initial state distribution. Thus,

$$\lim_{t \to \infty} \Pr(\hat{s}_y^{(t)} = s_y) = \lim_{t \to \infty} \Pr(s_y^{(t)} = s_y) \quad \forall s_y \in \mathcal{S}_y \tag{24}$$

Finally putting these results together, the following equalities hold for $(s_y, a_y, s'_y) \in \{(s_y, a_y, s'_y) | O_{\mathcal{M}_y}(s_y, a_y) = 1, s_y, s'_y \in \mathcal{S}_y, a_y \in \mathcal{A}_y\}$

$$\sigma_{\hat{\pi}_x}^{x \to y}(s_y, a_y, s'_y) \overset{\text{Lemma 5}}{=} \lim_{t \to \infty} \Pr(\hat{s}_y^{(t)} = s_y, \hat{a}_y^{(t)} = a_y, \hat{s}_y^{(t+1)} = s'_y; \mathcal{P})$$

$$= \lim_{t \to \infty} \Pr(\hat{s}_y^{(t)} = s_y) \Pr(\hat{a}_y^{(t)} = a_y | \hat{s}_y^{(t)} = s_y) \Pr(\hat{s}_y^{(t+1)} = s'_y | \hat{s}_y^{(t)} = s_y, \hat{a}_y^{(t)} = a_y)$$

$$\overset{\text{Eq (21),(22)}}{=} \lim_{t \to \infty} \Pr(\hat{s}_y^{(t)} = s_y) \Pr(a_y^{(t)} = a_y | s_y^{(t)} = s_y) \Pr(s_y^{(t+1)} = s'_y | s_y^{(t)} = s_y, a_y^{(t)} = a_y)$$

$$= \pi_y(a_y | s_y) \mathbb{1}(s'_y = P_y(s_y, a_y)) \lim_{t \to \infty} \Pr(\hat{s}_y^{(t)} = s_y)$$

$$\overset{\text{Eq (24)}}{=} \pi_y(a_y | s_y) \mathbb{1}(s'_y = P_y(s_y, a_y)) \lim_{t \to \infty} \Pr(s_y^{(t)} = s_y)$$

$$= \lim_{t \to \infty} \Pr(s_y^{(t)} = s_y) \Pr(a_y^{(t)} = a_y | s_y^{(t)} = s_y) \Pr(s_y^{(t+1)} = s'_y | s_y^{(t)} = s_y, a_y^{(t)} = a_y)$$

$$\overset{\text{Lemma 5}}{=} \sigma_{\pi_y}^y(s_y, a_y, s'_y)$$

The constant terms are moved in and out of the limit in the fourth and sixth line since the stationary distribution over states exist as $\mathcal{M}_y$ is unichain and $\pi_y$ is optimal for $\mathcal{M}_y$. This allows us to conclude that $\sigma_{\hat{\pi}_x}^{x \to y} = \sigma_{\pi_y}^y$ since $\sigma_{\pi_y}^y$ is supported on $\{(s_y, a_y, s'_y) | O_{\mathcal{M}_y}(s_y, a_y) = 1, s_y, s'_y \in \mathcal{S}_y, a_y \in \mathcal{A}_y\}$ by Lemma 6.

Now we show the ($\Leftarrow$) direction. We first introduce some overloaded notation:

$$\sigma_{\hat{\pi}_x}^x(s_x) = \lim_{T \to \infty} \frac{1}{T} \sum_{t=0}^{T-1} \Pr(s_x^{(t)} = s_x; \hat{\pi}_x, P_x, \eta_x) \overset{\text{Lemma5}}{=} \lim_{t \to \infty} \Pr(s_x^{(t)} = s_x; \hat{\pi}_x, P_x, \eta_x)$$

$$\sigma_{\hat{\pi}_x}^x(s_x, a_x) = \lim_{T \to \infty} \frac{1}{T} \sum_{t=0}^{T-1} \Pr(s_x^{(t)} = s_x, a_x^{(t)} = a_x; \hat{\pi}_x, P_x, \eta_x)$$

$$\overset{\text{Lemma 5}}{=} \lim_{t \to \infty} \Pr(s_x^{(t)} = s_x, a_x^{(t)} = a_x; \hat{\pi}_x, P_x, \eta_x)$$

$$= \lim_{t \to \infty} \Pr(s_x^{(t)} = s_x; \hat{\pi}_x, P_x, \eta_x) \hat{\pi}_x(a_x | s_x)$$

$$= \hat{\pi}_x(a_x | s_x) \lim_{t \to \infty} \Pr(s_x^{(t)} = s_x; \hat{\pi}_x, P_x, \eta_x)$$

$$= \hat{\pi}_x(a_x | s_x) \sigma_{\hat{\pi}_x}^x(s_x) \tag{25}$$

Then,

$$\sigma_{\hat{\pi}_x}^x(s_x, a_x, s_x') \overset{\text{Lemma 5}}{=} \lim_{t \to \infty} \Pr(s_x^{(t)} = s_x, a_x^{(t)} = a_x, s_x^{(t+1)} = s_x'; \hat{\pi}_x, P_x, \eta_x)$$

$$= \lim_{t \to \infty} \Pr(s_x^{(t+1)} = s_x' | s_x^{(t)} = s_x, a_x^{(t)} = a_x) \Pr(s_x^{(t)} = s_x, a_x^{(t)} = a_x; \hat{\pi}_x, P_x, \eta_x)$$

$$= \mathbb{1}(s_x' = P_x(s_x, a_x)) \lim_{t \to \infty} \Pr(s_x^{(t)} = s_x, a_x^{(t)} = a_x; \hat{\pi}_x, P_x, \eta_x)$$

$$= \mathbb{1}(s_x' = P_x(s_x, a_x)) \sigma_{\hat{\pi}_x}^x(s_x, a_x) \tag{26}$$

$$= \mathbb{1}(s_x' = P_x(s_x, a_x)) \hat{\pi}_x(a_x | s_x) \sigma_{\hat{\pi}_x}^x(s_x) \tag{27}$$

Given $f, g$ that satisfy objective 1, 2, and 3 we construct $(\phi, \psi)$ as follows and show that $(\phi, \psi) \in \Gamma(\mathcal{M}_x, \mathcal{M}_y)$:

$$\phi(s_x) = \begin{cases} f(s_x) & \text{if } s_x \in \text{supp}(\sigma_{\hat{\pi}_x}^x(s_x)) \\ s_y^d & \text{otherwise} \end{cases}$$

$$\psi(a_x) = \begin{cases} g^{-1}(a_x) & \text{if } a_x \in \mathcal{A}_{\hat{\pi}_x} = \bigcup_{s_x \in \mathcal{S}_x} \text{supp}(\hat{\pi}_x(\cdot | s_x)) \\ a_y^d & \text{otherwise} \end{cases}$$

where $s_y^d, a_y^d$ are dummy state, actions such that $O_{\mathcal{M}_y}(s_y^d, a_y) = 0 \ \forall a_y \in \mathcal{A}_y$ and $O_{\mathcal{M}_y}(s_y, a_y^d) = 0 \ \forall s_y \in \mathcal{S}_y$. Such dummy state, actions always exist per Assumption 1. Mapping to the dummy state, action will ensure that the constructions will not map suboptimal state, action pairs from domain $x$ to optimal state action pairs in domain $y$. The following statement holds for our construction $(\phi, \psi)$:

$$(s_x^*, a_x^*) \in \text{supp}(\sigma_{\hat{\pi}_x}^x(s_x, a_x)) \iff O_{\mathcal{M}_y}(\phi(s_x^*), \psi(a_x^*)) = 1 \quad \forall s_x^* \in \mathcal{S}_x, a_x^* \in \mathcal{A}_x \tag{28}$$

We first prove the forward direction: $(s_x^*, a_x^*) \in \text{supp}(\sigma_{\hat{\pi}_x}^x(s_x, a_x)) \Rightarrow \sigma_{\hat{\pi}_x}^x(s_x^*, a_x^*) \overset{\text{Eq25}}{=} \sigma_{\hat{\pi}_x}^x(s_x^*) \hat{\pi}_x(a_x^* | s_x^*) > 0$, so $\sigma_{\hat{\pi}_x}^x(s_x^*) > 0$, i.e $s_x^* \in \text{supp}(\sigma_{\hat{\pi}_x}^x(s_x))$, and $\hat{\pi}_x(a_x^* | s_x^*) > 0$. Furthermore, $\hat{\pi}_x(a_x^* | s_x^*) > 0 \Rightarrow g^{-1}(a_x^*) \in \text{supp}(\pi_y(\cdot | f(s_x^*)))$ since $g$ is injective. To see this, assume $\exists (s_x^*, a_x^*)$ such that $\hat{\pi}_x(a_x^* | s_x^*) > 0$ but $g^{-1}(a_x^*) \notin \text{supp}(\pi_y(\cdot | f(s_x^*)))$. Then there must exists $a_y' \in \text{supp}(\pi_y(\cdot | f(s_x^*)))$ such that $a_y' \neq g^{-1}(a_x^*)$ but $g(a_y') = g(g^{-1}(a_x^*)) = a_x^*$ contradicting the injectivity of $g$ on $\mathcal{A}_y$. Putting these results together we obtain $\psi(a_x^*) = g^{-1}(a_x^*) \in \text{supp}(\pi_y(\cdot | \phi(s_x^*)))$. Since $\pi_y$ is a stochastic optimal policy and $\mathcal{M}_y$ is unichain, $\psi(a_x^*) \in \text{supp}(\pi_y(\cdot | \phi(s_x^*))) \Rightarrow O_{\mathcal{M}_y}(\phi(s_x^*), \psi(a_x^*)) = 1$ by Corollary 1.

For the converse direction we prove the contrapostive: $(s_x^*, a_x^*) \notin \text{supp}(\sigma_{\hat{\pi}_x}^x(s_x, a_x)) \Rightarrow O_{\mathcal{M}_y}(\phi(s_x^*), \psi(a_x^*)) = 0 \ \forall s_x \in \mathcal{S}_x, a_x \in \mathcal{A}_x$. We exhaustively consider all cases in which $(s_x^*, a_x^*) \notin \text{supp}(\sigma_{\hat{\pi}_x}^x(s_x, a_x))$, i.e $\sigma_{\hat{\pi}_x}^x(s_x^*, a_x^*) \overset{\text{Eq25}}{=} \hat{\pi}_x(a_x^* | s_x^*) \sigma_{\hat{\pi}_x}^x(s_x^*) = 0$. If $\sigma_{\hat{\pi}_x}^x(s_x^*) = 0$, then $s_x^* \notin \text{supp}(\sigma_{\hat{\pi}_x}^x(s_x))$, so $O_{\mathcal{M}_y}(\phi(s_x^*), a_y) = O_{\mathcal{M}_y}(s_y^d, a_y) = 0 \ \forall a_y \in \mathcal{A}_y$. Else if $\hat{\pi}_x(a_x^* | s_x^*) = 0, \sigma_{\hat{\pi}_x}^x(s_x^*) > 0$ and $a_x \notin \mathcal{A}_{\hat{\pi}_x}$ then $O_{\mathcal{M}_y}(s_y, \psi(a_x^*)) = O_{\mathcal{M}_y}(s_y, a_y^d) = 0 \ \forall s_y \in \mathcal{S}_y$. Finally, consider the case $\hat{\pi}_x(a_x^* | s_x^*) = 0, \sigma_{\hat{\pi}_x}^x(s_x^*) > 0$ and $a_x^* \in \mathcal{A}_{\hat{\pi}_x}$. Assume for contradiction that $O_{\mathcal{M}_y}(\phi(s_x^*), \psi(a_x^*)) = 1$. Then, $\psi(a_x^*) \in \text{supp}(\pi_y(\cdot | \phi(s_x^*))$ since $\pi_y$ is a covering optimal policy from Definition 6, which implies $g^{-1}(a_x^*) \in \text{supp}(\pi_y(\cdot | f(s_x^*))$ since $\sigma_{\hat{\pi}_x}^x(s_x^*) > 0$ and $a_x^* \in \mathcal{A}_{\hat{\pi}_x}$. It

follows that $g(g^{-1}(a_x^*)) \in \text{supp}(g(\pi_y(\cdot|f(s_x^*)))) \Rightarrow a_x^* \in \text{supp}(\hat{\pi}_x(\cdot|s_x^*))$ since $\hat{\pi}_x(\cdot|s_x^*)$ is the pushforward measure $g(\pi_y(\cdot|f(s_x^*)))$. Then, $\sigma_{\hat{\pi}_x}^x(s_x^*, a_x^*) \stackrel{\text{Eq25}}{=} \sigma_{\hat{\pi}_x}^x(s_x^*)\hat{\pi}_x(a_x^*|s_x^*) > 0$, since $\hat{\pi}_x(a_x^*|s_x^*) > 0$ and $\sigma_{\hat{\pi}_x}^x(s_x^*) > 0$, which contradicts $(s_x^*, a_x^*) \notin \text{supp}(\sigma_{\hat{\pi}_x}^x(s_x, a_x))$. This concludes the proof of Equation 28.

We proceed to show that the optimal policy and dynamics preservation properties hold for our construction $(\phi, \psi)$.

● Optimal Policy (Equation 1): From the converse direction of the above subclaim and the optimality of $\hat{\pi}_x$ the result immediate follows:

$$O_{\mathcal{M}_y}(\phi(s_x^*), \psi(a_x^*)) = 1 \stackrel{\text{Eq28}}{\Rightarrow} (s_x^*, a_x^*) \in \text{supp}(\sigma_{\hat{\pi}_x}^x(s_x, a_x))$$
$$\stackrel{\text{Eq25}}{\Rightarrow} \hat{\pi}_x(a_x^*|s_x^*) > 0$$
$$\stackrel{\text{Cor1}}{\Rightarrow} O_{\mathcal{M}_x}(s_x^*, a_x^*) = 1 \quad \forall s_x^* \in \mathcal{S}_x, a_x^* \in \mathcal{A}_x$$

● Surjection (Equation 2): Assume for contradiction $\exists (s_y^*, a_y^*)$ such that $O_{\mathcal{M}_y}(s_y^*, a_y^*) = 1$, but $\phi^{-1}(s_y^*) = \emptyset$ or $\psi^{-1}(a_y^*) = \emptyset$. Since $O_{\mathcal{M}_y}(s_y^*, a_y^*) = 1$ we have $s_y^* \neq s_y^d, a_y^* \neq a_y^d$. Thus $\phi(s_y^*)^{-1} = f^{-1}(s_y^*)$ and $\psi^{-1}(a_y^*) = \{(g^{-1})^{-1}(a_y^*)\} = \{g(a_y^*)\}$. Since $g$ is a function defined $\forall a_y \in \mathcal{A}_y$, it follows that $\psi^{-1}(a_y^*) \neq \emptyset$. Thus it must be that $\phi^{-1}(s_y^*) = \emptyset$. Let $s_y^{*'} = P_y(s_y^*, a_y^*)$. Then,

$$\sigma_{\hat{\pi}_x}^{x \to y}(s_y^*, a_y^*, s_y^{*'}) \stackrel{\text{Lemma 5}}{=} \lim_{t \to \infty} \Pr(\hat{s}_y^{(t)} = s_y^*, \hat{a}_y^{(t)} = a_y^*, \hat{s}_y^{(t+1)} = s_y^{*'})$$

$$= \lim_{t \to \infty} \Pr(\hat{s}_y^{(t+1)} = s_y^{*'}|\hat{s}_y^{(t)} = s_y^*, \hat{a}_y^{(t)} = a_y^*) \Pr(\hat{s}_y^{(t)} = s_y^*, \hat{a}_y^{(t)} = a_y^*)$$

$$= \lim_{t \to \infty} \Pr(\hat{s}_y^{(t+1)} = s_y^{*'}|\hat{s}_y^{(t)} = s_y^*, \hat{a}_y^{(t)} = a_y^*) \Pr(\hat{a}_y^{(t)} = a_y^*|\hat{s}_y^{(t)} = s_y^*) \Pr(\hat{s}_y^{(t)} = s_y^*)$$

$$= \lim_{t \to \infty} \Pr(\hat{s}_y^{(t+1)} = s_y^{*'}|\hat{s}_y^{(t)} = s_y^*, \hat{a}_y^{(t)} = a_y^*)\pi_y(a_y^*|s_y^*) \sum_{s_x \in \phi^{-1}(s_y^*)} \Pr(\hat{s}_x^{(t)} = s_x)$$

$$= \lim_{t \to \infty} \Pr(\hat{s}_y^{(t+1)} = s_y^{*'}|\hat{s}_y^{(t)} = s_y^*, \hat{a}_y^{(t)} = a_y^*)\pi_y(a_y^*|s_y^*) \cdot 0$$

$$= 0$$

However,

$$\sigma_{\pi_y}^y(s_y^*, a_y^*, s_y^{*'}) \stackrel{\text{Eq 27}}{=} \mathbb{1}(P_y(s_y^*, a_y^*) = P_y(s_y^*, a_y^*))\pi_y(a_y^*|s_y^*)\sigma_{\pi_y}^y(s_y^*)$$

$$= \pi_y(a_y^*|s_y^*)\sigma_{\pi_y}^y(s_y^*) > 0$$

To see why the last inequality holds, first recall that $\mathcal{M}_y$ is unichain and $\pi_y$ is stochastic optimal for $\mathcal{M}_y$, so the stationary distribution over states have full support over $\mathcal{S}_y$ ($\because$ stationary distributions of irreducible markov chains are fully supported over the entire state space) Therefore $\sigma_{\pi_y}^y(s_y) \stackrel{\text{Lemma5}}{=} \lim_{t \to \infty} \Pr(s_y^{(t)} = s_y; \pi_y, P_y) > 0 \quad \forall s_y \in \mathcal{S}_y$. Thus, we have $\sigma_{\pi_y}^y(s_y^*) > 0$. Furthermore, $\pi_y(a_y^*|s_y^*) > 0$ by Corollary 1. Putting these two results together, we obtain $\sigma_{\pi_y}^y(s_y^*, a_y^*, s_y^{*'}) > 0$. Then, $\sigma_{\hat{\pi}_x}^{x \to y} \neq \sigma_{\pi_y}^y$ which contradicts the satisfiability of objective 3.

● Dynamics (Equation 3): Assume for contradiction that $\exists s_x^-, a_x^-$ and $s_x^{-'} = P_x(s_x^-, a_x^-)$ such that $O_{\mathcal{M}_y}(\phi(s_x^-), \psi(a_x^-)) = 1$ but the dynamics preservation property is violated, i.e $P_y(\phi(s_x^-), \psi(a_x^-)) \neq \phi(P_x(s_x^-, a_x^-)) = \phi(s_x^{-'})$. If $(s_x^-, a_x^-) \notin \text{supp}(\sigma_{\hat{\pi}_x}^x(s_x, a_x))$, then $O_{\mathcal{M}_y}(\phi(s_x^-), \psi(a_x^-)) = 0$ by Equation 28 which contradicts $O(\phi(s_x^-), \psi(a_x^-)) = 1$. Thus, it must be that $(s_x^-, a_x^-) \in \text{supp}(\sigma_{\hat{\pi}_x}^x(s_x, a_x))$ which further implies $(s_x^-, a_x^-, s_x^{-'}) \in \text{supp}(\sigma_{\hat{\pi}_x}^x(s_x, a_x, s_x'))$ by Equation 26 and $\phi(s_x^-) = f(s_x^-), \psi(a_x^-) = g^{-1}(a_x^-)$ by Equation 25 since $\sigma_{\hat{\pi}_x}^x(s_x^-) > 0, \hat{\pi}(a_x^-|s_x^-) > 0$.

Let $\mathcal{F} : \mathcal{S}_x \times g(\mathcal{A}_y) \times \mathcal{S}_x \to \mathcal{S}_y \times \mathcal{A}_y \times \mathcal{S}_y$ be a function $(a, b, c) \mapsto (f(a), g^{-1}(b), f(c))$. Then, by Lemma 7, we have $\sigma_{\hat{\pi}_x}^{x \to y}(s_x, a_x, s_x') = \mathcal{F}(\rho_{\hat{\pi}_x}^x(s_x, a_x, s_x')) \stackrel{\text{Lemma5}}{=} \mathcal{F}(\sigma_{\hat{\pi}_x}^x(s_x, a_x, s_x'))$. So,

$$\sigma_{\hat{\pi}_x}^x(s_x^-, a_x^-, s_x^{-'}) > 0 \Rightarrow \sigma_{\hat{\pi}_x}^{x \to y}(\mathcal{F}(s_x^-, a_x^-, s_x^{-'})) = \sigma_{\hat{\pi}_x}^{x \to y}(f(s_x^-), g^{-1}(a_x^-), f(s_x^{-'})) > 0$$

Thus, $(f(s_x^-), g^{-1}(a_x^-), f(s_x^{-'})) = (\phi(s_x^-), \psi(a_x^-), \phi(s_x^{-'})) \in \text{supp}(\sigma_{\hat{\pi}_x}^{x \to y}(s_x, a_x, s_x'))$. However,

$$\sigma_{\pi_y}^y(\phi(s_x^-), \psi(a_x^-), \phi(s_x^{-'})) \overset{\text{Eq 26}}{=} \sigma_{\pi_y}^y(\phi(s_x^-), \psi(a_x^-)) \mathbb{1}\big(\phi(s_x^{-'}) = P_y(\phi(s_x^-), \psi(a_x^-))\big)$$

$$= \sigma_{\pi_y}^y(\phi(s_x^-), \psi(a_x^-)) \cdot 0$$

$$= 0$$

Thus, $\text{supp}(\sigma_{\hat{\pi}_x}^{x \to y}) \neq \text{supp}(\sigma_{\pi_y}^y) \Rightarrow \sigma_{\hat{\pi}_x}^{x \to y} \neq \sigma_{\pi_y}^y$ which contradicts $f, g$ satisfying objective 3. This concludes the proof of the main theorem.

$\square$

