# OpenReview forum: "Cross Domain Imitation Learning"
_ICLR.cc/2020/Conference — Reject_

### Official Review · AnonReviewer2 · 2019-10-22
**Official Blind Review #2**

**Rating:** 3

**Review:**

Summary
This paper proposes a method to perform alignment between demonstrations in different domains in order to adapt a policy from expert domain to the test domain. They also include a formalism that studies alignments of different MDPs.

Strengths
1) The paper presents a formal framework to study many recent methods in cross-domain/cross-view imitation learning with a common lens.
2) The solution proposed by the authors might have a big practical advantage for cross-domain imitation learning settings. They claim that they do not need to learn the policy using RL in the new domain as they learn the action mapping between the two domains to retrieve the corresponding action in new domain based on the action in expert domain. This is a very interesting result.
3) The solution is quite modular with 3 important parts: i) state mapping network, ii) action mapping network, iii) dynamics model.

Weakness
The biggest weakness of the paper is in the evaluation section.
1) Issues with baselines:
i) There seems to be some problem with UMA. In Figure 4, it cannot figure out the alignment with self domain. Why is that the case? Where is the cartpole in UMA row for pen-cartpole alignment ? Given such poor performance during alignment is it fair to compare to UMA in the subsequent section?
ii) Why are the other methods not able to align well when there is no domain shift?
iii) How is dynamic time warping actually used to calculate the state correspondences for training IfO and IF? Is it calculated on top of the states?
iv) How are the policies trained for the transferability task? Which RL algorithm? "Baselines fail to learn the writing task as an inaccurate proxy reward function harms performance."
2 aspects are involved here:
1) initializing from alignment network
2) using rewards from the state representation to train the policy.
Do the algorithms still fail if the true reward is used with the alignment initialization?

Reward scaling is known to be important for RL algorithms[1]. May be features from these algorithms have to be scaled to scale the rewards for the RL algorithms to work.

2) Issues with environments:
i) The environments considered for domain transfer seem quite trivial for the alignment between them to be a big problem. For example, the authors present the task R2W: reacher2-tp that has a "third person" state space with a 180 camera angle offset. I am not sure how this rotation changes the state for the policy that uses states as input. One of the states of reacher is vector between goal and state which should still be informative enough to solve the task. Presumably the rotation creates a different view for the image based experiments. There also the transformation is quite easy for the spatial autoencoder given that it has access to coordinates through the spatial softmax layer.

ii) It is unclear what the R2W task entails or how difficult it is.

"The transfer task is writing letters as fast as possible. The transfer task differs from the alignment tasks in two key aspects: the end effector must draw a straight line from a letter’s vertex to vertex and minimally slow down at the vertices. "

It seems there is a term that penalizes the policy from doing things quickly. Some questions regarding this task:
1) How is this task specified?
2) Is agent rewarded for slowing down too?  Does the goal of the reacher and the state representation corresponding to the vector between end effector and goal get updated once it reached a particular point? Does the agent have to write one particular letter? Why do the other baselines not work at all on this task?

iii) Is it possible to run experiments on environments presented in Invariant Features paper so that the importance of CDIL can be assessed better.

4) "HIGH-LEVEL COMPARISON TO BASELINES" section is misleading. While the different baselines (TCN, IfO, TPIL) were developed in the context of view mismatch that does not mean the algorithm cannot be applied for embodiment mismatch. For IF also the same idea can be applied for viewpoint mismatch. TCN also proposed single-view version which can be used to learn representations without paired data. Hence, it is not right to say TCN can't work with unpaired alignment data. It also needs to be stated in this section that CDIL requires access to actions while some baselines like TCN do not need actions that can be used for training.

Questions
1) How were the alignment videos generated?

Decision
This paper has the potential to be an important paper in this field. But at this point needs further empirical evaluation with stronger baselines and known benchmark environments.

Minor Comments:
1)  "boltmzman machine reconstruction error" - Boltzmann

References
[1] "Deep Reinforcement Learning That Matters." Peter Henderson, Riashat Islam, Philip Bachman, Joelle Pineau, Doina Precup, David Meger

**Experience Assessment:**

I have published one or two papers in this area.

**Review Assessment: Checking Correctness Of Derivations And Theory:**

I assessed the sensibility of the derivations and theory.

**Review Assessment: Checking Correctness Of Experiments:**

I carefully checked the experiments.

**Review Assessment: Thoroughness In Paper Reading:**

I read the paper thoroughly.

---

> ### Author Response · Authors · 2019-11-08
> **Reply to Reviewer #2, part 1**
>
> We thank the reviewer for the detailed comments. We respond to your comments/questions below. We have also uploaded a revised draft addressing all suggested improvements. We’re happy to add more experiments if the reviewer has suggestions.
>
>
> RESPONSE TO REVIEWER COMMENTS/QUESTIONS
>
> Q: Where is the cartpole in Figure 4 UMA? Fair to compare in subsequent sections?
>
> A: UMA outputs cartpole coordinates which are completely out of bounds, hence the cart pole cannot be seen. Let $X$ be the state space of cartpole domain, $Y$ be the state space of pendulum domain and $E$ be a common feature space. UMA targets to learn linear maps (projections) $F: X \rightarrow E$ and $G: Y \rightarrow E$ such that if two states $x, y$ have similar local geometry, then their embeddings are close, i.e $Fx \approx Gy$ if $R_x$ is ‘similar’ to $R_y$ where $R_x, R_y$ are the local geometry matrices of state $x$ and state $y$, respectively. Approximately 50$\%$ of the demonstration states for domain $Y$ (pendulum) are near one state $y^*$ (i.e. pendulum standing upright) and thus $R_{y^*}$ is close to a zero matrix. Based on the local geometry similarity calculation of UMA, the distance $d(x, y)$ between two states $x, y$ satisfies $d(x,y) <= 2 \min \{ ||R_x||_F, ||R_y||_F \}$. Thus, all the states in domain $X$ are considered similar to state $y^* \in Y$ since $||R_{y*}||_F \approx 0$. In order to minimize the cost defined in the UMA algorithm, $F$ should map all the states in $X$ to one embedding $e^* \in E$ corresponding to the embedding for $y^*$. The only way to achieve this is to let $F$ be a near-zero matrix, which will make $||F^{-1}||$ very large. Moreover, a smaller fraction of states in $X$ domain demonstrations are near one point. Thus $||G||$ is not small enough to neutralize the effect of $F^{-1}$ and the pendulum->cartpole mapping $F^{-1}G$ has a very large norm. This causes points mapped points to be out of bounds. Comparisons to UMA in the subsequent sections show that obtaining proper state alignments is important for attaining good CDIL performance. We’ve added a summary of why UMA fails to the end of section 6.1.
>
>
> Q: Why are the other methods not able to align well when there is no domain shift?
>
> A: In section 6.1 we stated the reason: “The key reason behind this performance gap is that most baselines (Gupta et al., 2017; Liu et al., 2018) obtain state maps from time-aligned demonstration data. However, the considered alignment task set contains unaligned demonstrations with diverse starting states, up to 2x differences in demonstration lengths, and varying task execution rates.” In summary, even when there is no domain shift, such as in pen<->pen, the trajectory data is unpaired and unaligned, making prior methods fail due to the time-alignment assumption.
>
>
> Q: How is dynamic time warping actually used to calculate the state correspondences for training IfO and IF?
>
> A: For IF, Dynamic Time Warping (DTW) uses the (learned) feature space as a metric space to estimate the domain correspondences. For IfO DTW is applied to the state space. We follow the EM procedure outlined in the IF paper: https://arxiv.org/pdf/1703.02949.pdf. We will also be releasing code that includes the full implementation including dynamic time warping. We added these descriptions to Appendix C, “obtaining state correspondences”.
>
>
> Q: How are the policies trained for the transferability task? Which RL algorithm?
>
> A: For the baselines, we first pretrain the policy on the transfer task with the proxy reward induced by the
> statemap obtained during the alignment phase. We then train the policy with the ground truth reward function. All RL steps are performed with DDPG. The transferability plots (Fig 5. Right) show the learning curve for the second phase where the pretrained policy is trained on the ground truth reward with DDPG.
> We have added this description added to Appendix C, Transfer Learning.
>
>
> Q. Do the algorithms still fail if the true reward is used with the alignment initialization?
>
> A: Could the reviewer please clarify what they mean by “alignment initialization”? The transferability plot is showing the learning curve on training with the true reward after pretraining on the proxy reward defined by the learned state map. This is in agreement with the “negative transfer” results in prior literature [1], e.g pretraining on Atari Pong leads to worse asymptotic performance on Atari Breakout. Mainly, training on an unrelated reward function (in this case a “wrong” reward function) can lead to asymptotically worse performance on the transfer task. Moreover, it’s not possible to directly use the learned alignments as policy initialization for the baselines since they do not learn an action map.

---

> ### Author Response · Authors · 2019-11-08
> **Reply to Reviewer #2, part 2**
>
>
> Q. Did you try reward scaling?
>
> A: Yes. We’ve also attempted standard RL optimization tricks (e.g observation normalization used in OpenAI baselines) and weight regularization make the baselines train to reach expert level performance and observed them failing due to negative transfer [1]. We can add the results of trying different optimization tricks to the final version.
>
>
> Q. The environments considered for domain transfer seem quite trivial for the alignment between them to be a big problem.
>
> A: The W2C task is on par with the difficulty of reaching tasks presented in the IF paper (https://arxiv.org/pdf/1703.02949.pdf), namely in that the reacher must mobilize to new locations to solve the target task. Our setting is more difficult in the sense that we seek to generalize to the target task without the ground truth reward or an additional RL step, while IF, UMA, CCA have all assumed access to the ground truth reward for the target task.
> The R2W task is similar to the reacher task used in the original TPIL paper (https://arxiv.org/abs/1703.01703), but with a significantly larger (10x larger) viewpoint mismatch making it more challenging. (albeit they don’t require an alignment tasks set)
> Both papers were published ICLR in the last 3 years. Furthermore the additional main challenge of our experimental settings is that data is given in the form of “unpaired unaligned” demonstrations. This was sufficient to demonstrate the limitations of the time-alignment assumptions made in prior work.
>
>
> Q. What is challenging about the Third Person environment? One of the states of reacher is vector between goal and state which should still be informative enough to solve the task. Presumably the rotation creates a different view for the image based experiments. There also the transformation is quite easy for the spatial autoencoder given that it has access to coordinates through the spatial softmax layer.
>
> A: We clarify the challenges of the third person environment:
> First when using the robot’s internal state space: a reacher joint configuration $(\theta_1, \theta_2)$ corresponds to $(\theta_1 + \alpha, \theta_2 + \alpha)$ in the third person domain with a rotation of $\alpha$ degrees along the $z$-axis (the reacher lies on the $x,y$ plane) while the goal coordinates $(x, y)$ stay the same. The challenge during the alignment phase is to learn what the value of $\alpha$ is. Learning this from unpaired, unaligned data is challenging as we see that the baselines fail, even with dynamic time warping. If one were to apply the same policy learned in the original domain to the third person domain it would completely fail. For example, let $(\theta_1, \theta_2)$ correspond to the original reacher states for which the end effector reaches goal $(x, y)$. Then the optimal policy $\pi^*$ will output $\pi^*(\theta_1, \theta_2) = (0, 0)$, i.e we add no torque to the joints and keep the reacher still at the goal. However, in the third person domain, $(\theta_1, \theta_2)$ corresponds to a state $\alpha$ degrees away from target configuration. So the third person reacher would not be moving in a state that has not yet reached the goal. The difference vector between the end-effector and goal was removed from the state space to make the task more challenging, so it cannot be used. For these reasons we see that the pretrained baselines (train the policy on the alignment task set MDPs for domain $x$, and directly apply that policy to the target task in domain $x$) also fail in Figure 5 (Bottom-Left).
> In the image space: as the reviewer pointed out, this creates a different view, e.g a reacher point down in the original domain is point up in the third person domain. Even if the spatial autoencoder is able to “perfectly” extract the reacher’s joint coordinates and its velocities, challenges would persist as described above. In practice we see that the spatial autoencoder extracts coordinates that are noisy, which poses an additional challenge for learning alignments. We have added more details about the third person environment in Appendix D.
>
>
> Q.  It is unclear what the R2W task entails or how difficult it is.
> Q-a ) How is this task specified?
> A: Vertices of the letter are spawned in sequence and the state representation contains the next target_vertex coordinate $(v_x, v_y)$. Here’s pseudocode for how the reward is calculated and how the next vertex location is updated:
> Let $e(s)$ be the location of the end effector for robot state $s$.
> if $||e(s) - (v_x, v_y)|| < \epsilon$: then $R(s) = 100$, update $(v_x, v_y) =$ next target vertex
> else : $R(s) = -1$
> In words: agent gets a large reward for hitting the vertex of a letter and the next vertex is spawned. Each step in the environment has a negative reward to promote fast task completion. The write task reward is significantly more sparse than the original reacher task reward which is defined as the inverse of the distance between the end effector and goal.

---

> ### Author Response · Authors · 2019-11-08
> **Reply to Reviewer 2, part 3**
>
>
> Q-b ) Is agent rewarded for slowing down too?
>
> A: No, the opposite is true. The agent is rewarded for performing the task as fast as possible. We have clarified the wording in section 6.2, “Reacher2Write”.
>
> Q-c ) Does the goal of the reacher and the state representation corresponding to the vector between end effector and goal get updated once it reached a particular point?
>
> A: The goal location in the state representation is updated to be the next vertex in the letter. However, the difference vector between end effector and goal is removed from the state to make the task more challenging. We’ve added more details about the writing task in Appendix D.
>
> Q-d ) Does the agent have to write one particular letter? Why do the other baselines not work at all on this task?
>
> A: Agents must write two letters “A”, “I” in sequence. This task is difficult due to the sparse reward (you only get positive reward upon hitting the goals unlike the original reacher environment which gives reward inversely proportional to the distance between the goal and end effector). Other baselines fail to learn good alignments between domains and as a result the proxy reward does more harm than good due to negative transfer [1]. We’ve added more details about the writing task in Appendix D. We will also include a more elaborate video of the environments and learned alignments in the final submission.
>
>
> Q. Is it possible to run experiments on environments presented in Invariant Features paper so that the importance of CDIL can be assessed better.
>
> A: The code for the exact environments used in the IF paper are not publicly available, but we have contacted the authors of the paper to see if we can obtain the environment source code and run experiments.
>
>
> Q. “HIGH-LEVEL COMPARISON TO BASELINES" section is misleading. Should add an “action accessibility” column.
>
> A: The high level comparison table has a checkpoint if the attribute was demonstrated in the paper. We agree that prior works (TCN, IfO, TPIL) have potential to be applied to the embodiment mismatch problem and would love to see works that attempt to do so. (TCN did show that you can learn interesting mappings between human-robot, but did not demonstrate that humans can teach new tasks to the robot using the mapping) We have added clarifications of the check/x marks meanings, a column for action accessibility, and interpretation of Table 2 in Appendix A.
>
>
> Q. How were the alignment videos generated?
>
> A: We rollout the imitator’s policy while applying the (learned) statemap to the visited internal states of the robot to obtain the corresponding expert domain states. These states are mapped to images by using the set_state() function in the mujoco environments.
>
>
> Q. This paper has the potential to be an important paper in this field. But at this point needs further empirical evaluation with stronger baselines and known benchmark environments.
>
> A: We believe we’ve done a thorough comparison against state-of-the-art baselines in the field and our environments clearly demonstrated shortcomings of prior work while highlighting the advantage of our method: namely, learning from unpaired, unaligned data and zero-shot imitation of the target task. If the reviewer suggests any additional environments/baselines we are happy to run additional experiments.
>
>
> Minor Comments:
> 1)  "boltmzman machine reconstruction error" - Boltzmann
>
> A: Thank you for pointing this out. It’s been revised.
>
>
> QUESTIONS FOR THE REVIEWER:
>
> Q1. Your decision statement mentioned benchmark environments and stronger baselines.
> (1). Could you suggest what benchmark environments you’re referring to? Are there any additional experimental settings you would like us to evaluate our approach on?
> (2). What stronger baselines are you referring to? Are there any particular ones you would like us to add?
>
> Q2. Could you clarify what you meant by “alignment initialization”?
>
>
> REFERENCES
> [1] “Characterizing and Avoiding Negative Transfer”, Zirui Wang, Zihang Dai, Barnabás Póczos, Jaime Carbonell, arXiv:1811.09751
> [2] “Manifold Alignment without Correspondence”, Chang Wang, Sridhar Mahadevan, IJCAI 2009.

---

### Official Review · AnonReviewer3 · 2019-10-23
**Official Blind Review #3**

**Rating:** 6

**Review:**

This paper proposes Generative Adversarial MDP Alignment (GAMA) for imitation learning. Given a set of paired MDPs, GAMA learns a state mapping f and an action mapping g such that one MDP can be reduced to another. For a new test MDP pair (x,y) where expert demonstrations are available for y, GAMA can use f to map a state of x to a state of y, mimic the expert behavior, then use g to map the expert action back to an action in x. The reduction is theoretically motivated, the optimization is based on adversarial learning and finally, experiments on common Gym environments show that GAMA can achieve effective transfer.

Pros
- The writing is great and easy to follow
- The method is theoretically motivated
- Experiments prove effective

Cons
- The proposed method may not work well for complicated environments

(1) In the discussion after Def.4, given an alignment task set D_{x,y}, how do we know whether a common (w.r.t. all MDP pairs) reduction exists? In other words, how do we know that the MDP pairs are from the same equivalent class (joint alignable) in practice?

(2) The alignment needs to learn a lot of components: state mapping f, action mapping g, domain dynamics P^x. The domain dynamics can be difficult to learn for complicated environments, which may jeopardize the learning of f and g as a result because they depend on the accuracy of the learned dynamics. The experiment only uses the hidden representation for the image as input. Such lower-dimensional representation is not always available.

(3) Experiment:
- What happens for the UMA in Fig.4 top-right?
- Table 1 only has the results of three alignment tasks. How about the rest?
- What are the error bars in Table 1 (and also Fig.5)? Based on how many runs?

Minors:
- It is more common to use "state-action pair" instead of "state, action pair".
- Sec.6.1, task task exemplify

**Experience Assessment:**

I have read many papers in this area.

**Review Assessment: Checking Correctness Of Derivations And Theory:**

I assessed the sensibility of the derivations and theory.

**Review Assessment: Checking Correctness Of Experiments:**

I assessed the sensibility of the experiments.

**Review Assessment: Thoroughness In Paper Reading:**

I read the paper at least twice and used my best judgement in assessing the paper.

---

> ### Author Response · Authors · 2019-11-08
> **Reply to Reviewer #3**
>
>
> Thank you for your constructive feedback. Below we respond to your questions. We've uploaded a revised draft that addresses all of your suggestions for improvement.
>
>
> Q. In the discussion after Def.4, given an alignment task set D_{x,y}, how do we know whether a common reduction exists?
>
> A: Applying Theorem 1, if the training error of GAMA is 0 then the learned alignments are jointly align all MDP pairs in the alignment task set. Thus a practical test of whether the MDP pairs in the alignment task set D_{x, y} are jointly alignable would be whether or not the GAMA training loss can be minimized close to 0.
>
>
> Q. The domain dynamics can be difficult to learn for complicated environments, which may jeopardize the learning of f and g as a result because they depend on the accuracy of the learned dynamics. The experiment only uses the hidden representation for the image as input. Such lower-dimensional representation is not always available.
>
> A: Our hidden representations in the image experiments are learned from images using spatial autoencoders. This modular approach of learning hidden representations via self-supervised learning then learning dynamics or a control policies on top of the learned representations has been shown to be effective in both robotics [1] and video-game [2] domains. In general, we agree that learning a latent space for dynamics model learning in high dimensional spaces is indeed a difficult problem undergoing active research. Learning image-to-image statemaps is also certainly an interesting problem, yet is still challenging even in the non-sequential decision making setting. (e.g CycleGAN is notoriously hard to train [3]) An interesting future research direction would be to extend CDIL with image-to-image translation maps and visual dynamics models.
>
>
> Q. What happens for UMA in Fig.4 top-right?
>
> A: UMA outputs cartpole coordinates which are completely out of bounds, hence the cart pole cannot be seen. Let $X$ be the state space of cartpole domain, $Y$ be the state space of pendulum domain and $E$ be a common feature space. UMA targets to learn linear maps (projections) $F: X \rightarrow E$ and $G: Y \rightarrow E$ such that if two states $x, y$ have similar local geometry, then their embeddings are close, i.e $Fx \approx Gy$ if $R_x$ is ‘similar’ to $R_y$ where $R_x, R_y$ are the local geometry matrices of state $x$ and state $y$, respectively. Approximately 50$\%$ of the demonstration states for domain $Y$ (pendulum) are near one state $y^*$ (i.e. pendulum standing upright) and thus $R_{y^*}$ is close to a zero matrix. Based on the local geometry similarity calculation of UMA, the distance $d(x, y)$ between two states $x, y$ satisfies $d(x,y) <= 2 \min \{ ||R_x||_F, ||R_y||_F \}$. Thus, all the states in domain $X$ are considered similar to state $y^* \in Y$ since $||R_{y*}||_F \approx 0$. In order to minimize the cost defined in the UMA algorithm, $F$ should map all the states in $X$ to one embedding $e^* \in E$ corresponding to the embedding for $y^*$. The only way to achieve this is to let $F$ be a near-zero matrix, which will make $||F^{-1}||$ very large. Moreover, a smaller fraction of states in $X$ domain demonstrations are near one point. Thus $||G||$ is not small enough to neutralize the effect of $F^{-1}$ and the pendulum->cartpole mapping $F^{-1}G$ has a very large norm. This causes points mapped points to be out of bounds. We’ve added a summary of why UMA fails to the end of section 6.1.
>
>
> Q. Table 1 only has the results of three alignment tasks. How about the rest?
>
> A: Table 1 shows quantitative evaluations of the learned statemaps when a simple ground truth reduction (e.g permuation) exists between the domains. It is difficult to quantitatively evaluate learned statemaps between domains that aren’t perfectly alignable, e.g snake3<->snake4, since it is not clear what the ground truth statemap should be. We instead implicitly evaluate the quality of more complex alignments by assessing how useful the alignments are for CDIL in section 6.2, 6.3.
>
>
> Q. What are the error bars in Table 1 (and also Fig.5)? Based on how many runs?
>
> A: Error bars/regions in Table 1/ Fig. 5 are the standard deviations obtained from training the alignment maps with multiple seeds. The randomness comes from (1). Sampling the alignment task set. (2). Optimization of GAMA’s training objective via SGD. All experiments were run with 5 seeds. We’ve clarified this in the captions for Table 1 and Fig. 5 in the uploaded revision.
>
>
> [1]. “Deep Spatial Autoencoders for Visuomotor Learning”, Chelsea Finn, Xin Yu Tan, Yan Duan, Trevor Darrell, Sergey Levine, Pieter Abbeel, arXiv:1509.06113
> [2]. “World Models”, David Ha, Jurgen Schmidhuber, arXiv:1803.10122
> [3] “Factors Affecting Accuracy in Image Translation based on Generative Adversarial Network”, Fumiya Yamashita, Ryohei Orihara, Yuichi Sei, Yasuyuki Tahara and Akihiko Ohsuga, ICAART 2018

---

### Official Review · AnonReviewer1 · 2019-10-28
**Official Blind Review #1**

**Rating:** 8

**Review:**

The paper proposes a learning approach for zero-shot imitation learning in an RL setting across domains with different embodiments and viewpoint mismatch. The proposed approach involves two steps, alignment and adaptation. In contrast to previous work, the alignment between domains, represented as MDPs, is learned from unpaired, unaligned samples from both domains. The paper presents a theoretical formulation of the cross-domain imitation learning problem, and presents an algorithm for training alignment and adaptation from data.

I think the paper is well written and theoretically well-founded. The authors provide experiments comparing to many previous works in cross-domain imitation learning and show that their approach outperforms previous approaches.

I only have minor comments.

1. For imitation between reacher2 and reacher3, there are multiple correspondences between the two domains due to the redundancy in the 3-link robot (and in general with n-link robots). How does GAMA deal with these redundancies? For example, an n-link robot is able to perform the same task with different configurations (elbow up, elbow down, etc.,) and all these will correspond to one configuration of a 2-link robot.

2. In practise, can 'alignment complexity' and 'adaptation complexity' help identify if transfer with GAMA between two domains is not beneficial? Results in Figure 5 only shows cases for GAMA in which the two metrics are good, resulting in good transfer. However, I am interested in knowing if there could be cases in which the two complexity metrics tell beforehand that transfer will not be beneficial.

3. In Figure 5's caption, Adaptation complexity is on Left and Alignment Complexity in the Middle. However, the text refers to Alignment complexity on the Left and Adaptation complexity in the Middle.

4. A reference is missing in Appendix B.

**Experience Assessment:**

I have read many papers in this area.

**Review Assessment: Checking Correctness Of Derivations And Theory:**

I assessed the sensibility of the derivations and theory.

**Review Assessment: Checking Correctness Of Experiments:**

I assessed the sensibility of the experiments.

**Review Assessment: Thoroughness In Paper Reading:**

I read the paper at least twice and used my best judgement in assessing the paper.

---

> ### Author Response · Authors · 2019-11-08
> **Reply to Reviewer #1**
>
>
> Thank you for your feedback. Below we respond to your questions. We've uploaded a revised draft that addresses all of your suggestions.
>
> Q. For imitation between reacher2 and reacher3, there are multiple correspondences between the two domains due to the redundancy in the 3-link robot (and in general with n-link robots). How does GAMA deal with these redundancies? For example, an n-link robot is able to perform the same task with different configurations (elbow up, elbow down, etc.,) and all these will correspond to one configuration of a 2-link robot.
>
> A: If there exists multiple MDP reductions from a lower dimensional robot to a high dimensional robot, any one of them will be a global optimum of our optimization objective by Theorem 1. Hence GAMA should learn to map the states of the lower dimensional robot to states of the high dimensional robot which together compose one (out of potentially many) sequence of configurations in which the high dimensional robot accomplishes the task. We observe this empirically: reacher3 has many different sequences of states that correspond to successful goal reaching (due to redundancy). The (learned) statemap from reacher2 to reacher3 converges to one of them. Last paragraph of section 3 is also relevant to this point.
>
>
> Q. In practise, can 'alignment complexity' and 'adaptation complexity' help identify if transfer with GAMA between two domains is not beneficial? Results in Figure 5 only shows cases for GAMA in which the two metrics are good, resulting in good transfer. However, I am interested in knowing if there could be cases in which the two complexity metrics tell beforehand that transfer will not be beneficial.
>
> A: Yes, we believe good adaptation complexity and good adaptation complexity implies good transferability since good zero-shot imitation performance will “kick start” the policy getting rid of the burn-in phase. This is especially useful if the target task reward is sparse. If the adaptation complexity and alignment complexity are bad, we indeed observed that transferability is worse. For example, in the W2C task if we train on a small alignment task set (which has worse adaptation/alignment performance), then the policy trains slower on the target task. We can include these results in the final submission.
>
>
> Q. In Figure 5's caption, Adaptation complexity is on Left and Alignment Complexity in the Middle. However, the text refers to Alignment complexity on the Left and Adaptation complexity in the Middle.
>
> A: Thank you for catching this. We have updated the submission.
>
>
> Q. A reference is missing in Appendix B.
>
> A: Thank you for pointing this out. We have updated the submission.

---

### Author Response · Authors · 2019-09-27
**Videos of cross domain state alignments by Generative Adversarial MDP Alignment (GAMA)**

Video of the cross domain state alignments learned by Generative Adversarial MDP Alignment (GAMA).
GAMA learns these statemaps from unpaired, unaligned demonstrations without any additional supervision.

link: https://youtu.be/l0tc1JCN_1M

---

### Decision · Program_Chairs · 2019-12-19

**Decision:**

Reject

**Comment:**

The authors propose a novel approach for imitation learning in settings where demonstrations are unaligned with the task (e.g., differ in terms of state and action space). The proposed approach consists of alignment and adaptation steps and theoretical insights are provided on whether given MDPs can be aligned. Reviewers were positive about the ideas presented in the paper, and several requests for clarification were well addressed by the authors during the rebuttal phase. Key evaluation issues remained unresolved. In particular, it was unclear to what degree performance differences were purely caused by issues in alignment, and reviewers did not see sufficient evidence to support claims about performance on the full cross domain learning setting.